

# Subglacial permafrost dynamics and erosion inside subglacial channels driven by surface events in Svalbard

Andreas Alexander[1,2], Jaroslav Obu[1], Thomas V. Schuler[1], Andreas Kääb[1], and Hanne H. Christiansen[2]

[1]Department of Geosciences, University of Oslo, 0316 Oslo, Norway
[2]Department of Arctic Geology, The University Centre in Svalbard, 9171 Longyearbyen, Norway

**Correspondence:** Andreas Alexander (andreas.alexander@geo.uio.no)

**Abstract.** Cold glacier beds, i.e. where the ice is frozen to its base, are widespread in polar regions. Common theories state that stable permafrost should exist under glacier beds on shorter time scales, varying from years to decades. Presently, only a few direct measurements of both subglacial permafrost and the processes influencing its thermal regime exist. Here, we present subglacial permafrost and active layer measurements obtained from within the basal drainage systems of two cold-based glaciers on Svalbard during the summer melt season. Temperature observations were obtained from subglacial sediment that was accessed through the drainage systems of the two glaciers in the winters before. The temperature records cover the periods from spring to autumn in 2016 and 2019, at the glaciers Larsbreen and Tellbreen in central Svalbard, respectively. The ground temperature below Larsbreen indicates colder ground conditions, whereas the temperatures of the Tellbreen drainage system show considerably warmer conditions, close to the freezing point. We suggest the latter is due to the presence of liquid water all year round inside the Tellbreen drainage system. Both drainage systems investigated show an increase in subglacial sediment temperatures after the disappearance of snow bridges and the subsequent connection to surface meltwater supply at the start of the summer melt season. Temperature records show influence of sudden summer water supply events, when heavy melt and rain left their signatures on the thermal regime and the erosion of the glacier bed. Observed vertical erosion can reach up to 0.9 m per day at the base of basal drainage channels during summer. We also show that the thermal regime under the subglacial drainage systems is not stable during summer, but experiences several freeze-thaw cycles driven by weather events. Our results show the direct importance of heavy melt events and rain on the thermal regime of subglacial permafrost and the erosion of the glacier bed in the vicinity of subglacial drainage channels. Increased precipitation and surface melt, as expected for future climate, will therefore likely lead to increased degradation of subglacial permafrost, as well as higher subglacial erosion around the preferential hydrological paths. This in turn might have significant impacts on proglacial and fjord ecosystems due to increased sediment and nutrient input.

## 1 Introduction

Cold-based glaciers and polythermal glaciers with cold–based margins are common and widespread in the polar regions. 15% of the terrestrial parts land of the Northern Hemisphere (excluding water bodies and glaciers) are underlain by permafrost and the mountainous areas of the Southern Hemisphere, as well as the ice-free areas of Antarctica are known to host large



permafrost areas (e.g., Obu et al., 2019, 2020). Interactions between glaciers and permafrost have, however, received only little attention in scientific literature and subglacial permafrost is not included into the estimations of permafrost extent. The reason therefore is that subglacial permafrost was long believed to either be not persistent or to shutdown basal processes, which have relevance for ice dynamics and geomorphological processes (e.g., Etzelmüller and Hagen, 2005; Haeberli, 2005; Harris and
Murton, 2005; Waller et al., 2012).

Permafrost is generally thought to occur as thin layers in the periphery of cold-based glaciers. In addition permafrost is generally believed to occur under thicker ice masses for two main reasons: First, glaciers may have overriden existing permafrost during ice advance (Mathews and Mackay, 1960; Cutler et al., 2000). Second, permafrost might have formed during
deglaciation due to reduced basal shear stresses and ice thinning (e.g., Dyke, 1993; Sollid and Sørbel, 1994). The latter process occurs regularly on Svalbard where the cold base of polythermal glaciers increases due to ice thinning (e.g., Björnsson et al., 1996; Nuth et al., 2019). Subglacial permafrost can survive over very long time under temperate ice due to reduced ground temperature gradients as a result of ice advance (Etzelmüller and Hagen, 2005). Subglacial permafrost temperature measurements of -13°C were reported under 1375 m thick ice of the Greenland Ice Sheet at Camp Century (Herron and Langway,
1979). The permafrost is thereby thought to persist due to a combination of very low air temperatures together with downward advection from cold firn (Kleman and Borgström, 1994; Dyke, 1993; Waller et al., 2012). Waller et al. (2012) list three principal locations under larger ice masses, where permafrost can occur: marginal fringes with sub-freezing basal temperatures due to thin ice cover (Astakhov et al., 1996; Murton et al., 2004); central areas of ice masses with low air temperatures and cold firn advection to the base with minimal strain heating at the same time (Dyke, 1993; Marshall and Clark, 2002); areas of flow
divergence where cold ice is brought into contact with the bed, and ice thickness and basal shear stresses are reduced (Dyke, 1993; Kleman et al., 1999). Given those principal locations, the spatial extend of subglacial permafrost is, however, still poorly understood (Waller et al., 2012).

Subglacial erosion under cold-based glaciers has been studied by Boulton (1972), who concluded that erosion below cold ice
could only occur due to englacially transported sediment in contact with the bed and due to quarrying and plucking (Boulton, 1972). High sedimentation rates in polythermal (e.g., Sollid and Sørbel, 1994; Hallet et al., 1996; Hodgkins et al., 2003) and cold-based glacier catchments (Etzelmüller et al., 2000) imply, however, additional sediment availability from dead-ice and till-dominated glacier forefields (Etzelmüller and Hagen, 2005). This suggests that further erosion mechanisms may exist, in addition to those described by Boulton (1972).


Field studies suggest that cold-based glaciers have active basal processes, including sliding (e.g., Echelmeyer and Zhongxiang, 1987; Cuffey et al., 1999), as well as substrate deformation at subfreezing temperatures (Fitzsismon et al., 1999) due to the transition from hard-frozen to plastic-frozen (Tsytovich, 1975; Waller et al., 2012). A better understanding of subglacial permafrost and related processes would be very beneficial, not only for a better understanding of the potential formation of new
permafrost, as glaciers recede due to warming, but also of glacier dynamics and geomorphological processes (Etzelmüller and



Hagen, 2005; Haeberli, 2005; Waller et al., 2012). The demand for direct measurements of the thermal conditions of subglacial sediment has long been posed (Menzies, 1981), but cold-based soft-bedded glaciers remain to date one of the most poorly understood types of glaciers (Waller et al., 2012).

In this study we take a first step towards overcoming this lack of understanding by providing direct temperature measurements from subglacial sediments under two different cold-based valley glaciers on Svalbard. We show the development of air temperatures as well as ice and sediment temperatures in and around subglacial channels during spring and the following melt season. Our measurements are further related to meteorological conditions. We present evidence for fast subglacial erosion inside the channels and provide a potential explanation.

## 2   Study sites and Methods

### 2.1   Field sites

The Svalbard archipelago extends between 74° and 80° N, and 10° and 35° E. Mean annual air temperatures in central Svalbard (Svalbard airport) were measured to be −5.9 °C between 1971 and 2000, with an increase of 1 °C per decade between 1971

and 2017, mostly due to rise in winter temperatures, making Svalbard one of the fastest warming regions (Hanssen-Bauer et al., 2019). The precipitation is generally low, with central Svalbard being the driest part of the archipelago. Mean annual precipitation was measured to be 196 mm between 1971 and 2000 at Svalbard airport, with a linear increase of 4% per decade between 1971 and 2017 and an autumn precipitation increase of 15% during the same period (Hanssen-Bauer et al., 2019). The annual rainfall is expected to increase by up to 65% from 1961-90 to 2071-2100 (Hanssen-Bauer et al., 2019), as well

as the frequency and intensity of heavy rainfall events until the end of the century (Isaksen et al., 2017). Given its climatic setting, about 59% of Svalbard's land area is covered by glaciers (Nuth et al., 2013). Most glaciers are polythermal, containing both, cold and temperate ice (Dallmann et al., 2015). Permafrost is traditionally considered as continuous over the entire archipelago, although recent modeling shows a degradation of near-surface permafrost in the coastal parts of western Svalbard (e.g., Etzelmüller et al., 2011; Obu et al., 2019).

   Two small cold-based valley glaciers are investigated in this study: Larsbreen and Tellbreen (Fig. 1). Larsbreen is a 3.1 km long land terminating valley glacier at 78.17° N and 15.5° E, located 1.3 km south of the main settlement Longyearbyen. It extends from 300 to 800 meter elevation above sea level on a north-facing slope, covers a surface area of 3 km$^2$, and has flow speeds of 1-4 m/yr (Etzelmüller et al., 2000). The glacier is entirely cold-based and has a maximum ice thickness of

around 120 meters (Alexander, 2017). It rests on easily erodible Eocene marine sandstones and shales of the Van Mijenfjorden Group (Dallmann et al., 2015). Two major lateral supraglacial streams enter into the glacier on the eastern and the western side respectively and reach the glacier bed in the frontal part, where they merge, before reemerging in the glacier forefield. The system explored in this study was accessed through a moulin cutting into a subglacial channel. The channel is ice-walled in

**Figure 1.** a) Overview map of Svalbard, showing the locations of Larsbreen and Tellbreen on the Svalbard archipelago. b) Frontal part of Tellbreen with the outline of the subglacial system indicated. Background aerial image ©Norwegian Polar Institute. c) Frontal part of Larsbreen with the center profile of the subglacial system. Background aerial image ©Norwegian Polar Institute.

its upper parts while completely eroded into the sedimentary bedrock (Nye-channel, i.e. a subglacial channel incised into the glacier bed) further downstream. It connects to the glacier terminus during summertime (map in figure 2).

Tellbreen, is a 3.5 kilometers long valley glacier, located at 78.13° N and 16.5 °E, 20 km to the northeast of Longyearbyen.
5    It covers an east-facing slope between 250 and 900 meters a.s.l.. The mean ice thickness is 59 meters and the area 2.8 km$^2$ (Bælum and Benn, 2011). The glacier is thought to be entirely cold-based even though year-round liquid water release from the subglacial system, which leads to an icing forming in front of the glacier in wintertime, is reported (Bælum and Benn, 2011). The bedrock geology consists of both Paleocene sandstones and shales from the Van Mijenfjorden Group, as well as middle Jurassic to lower Cretaceous sandstones and shales from the Adventdalen Group (Dallmann et al., 2015). Supraglacial channels



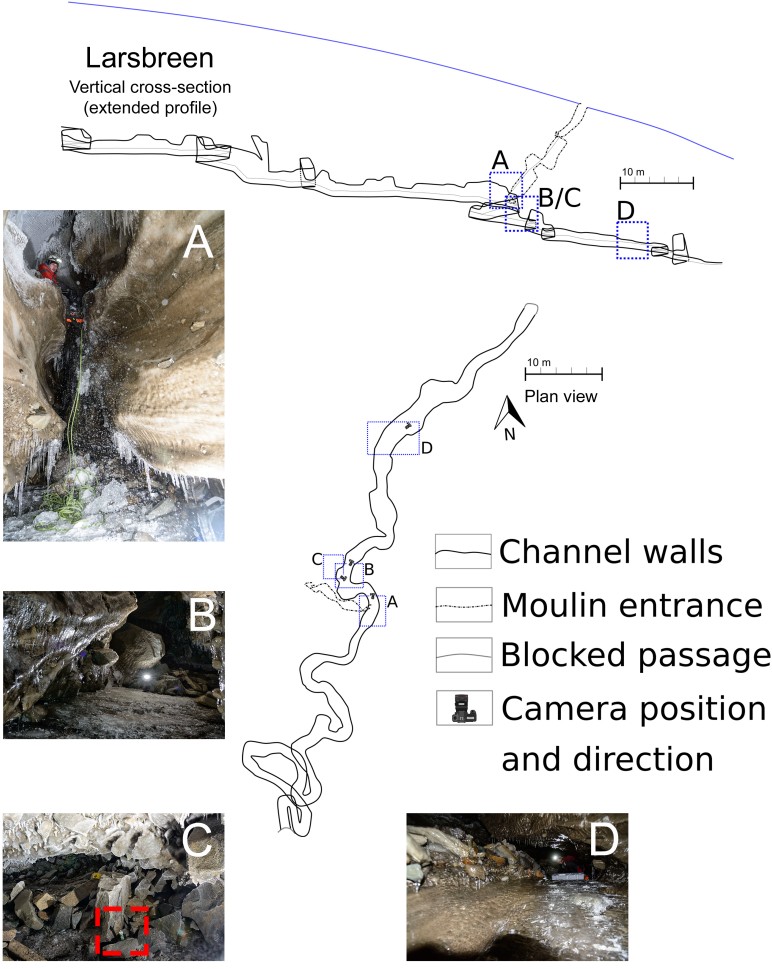

**Figure 2.** Plan view drainage system map of Larsbreen with picture locations (center) and vertical cross-section/ extended profile (upper sketch) with approximate glacier surface and photos. The drainage system could be entered by rappelling down a moulin/ channel, leading into the subglacial channel at (A). Channel looking upstream, picture taken from sensor location (B). Location of the sensor borehole at the side of the channel marked in red (C). Note that the logger box visible on the picture was later on moved higher up towards the ceiling of the drainage system. Low continuing passage towards the glacier front at (D). Dates of pictures: 7.11.2015 (A), 02.12.2015 (D), 05.03.2016 (B, C).

on Tellbreen exist on the southern and northern lateral sides of the glacier, and several englacial and subglacial channels can be found at the glacier tongue, with a subglacial system emerging at the southern side. The latter system was open during winter



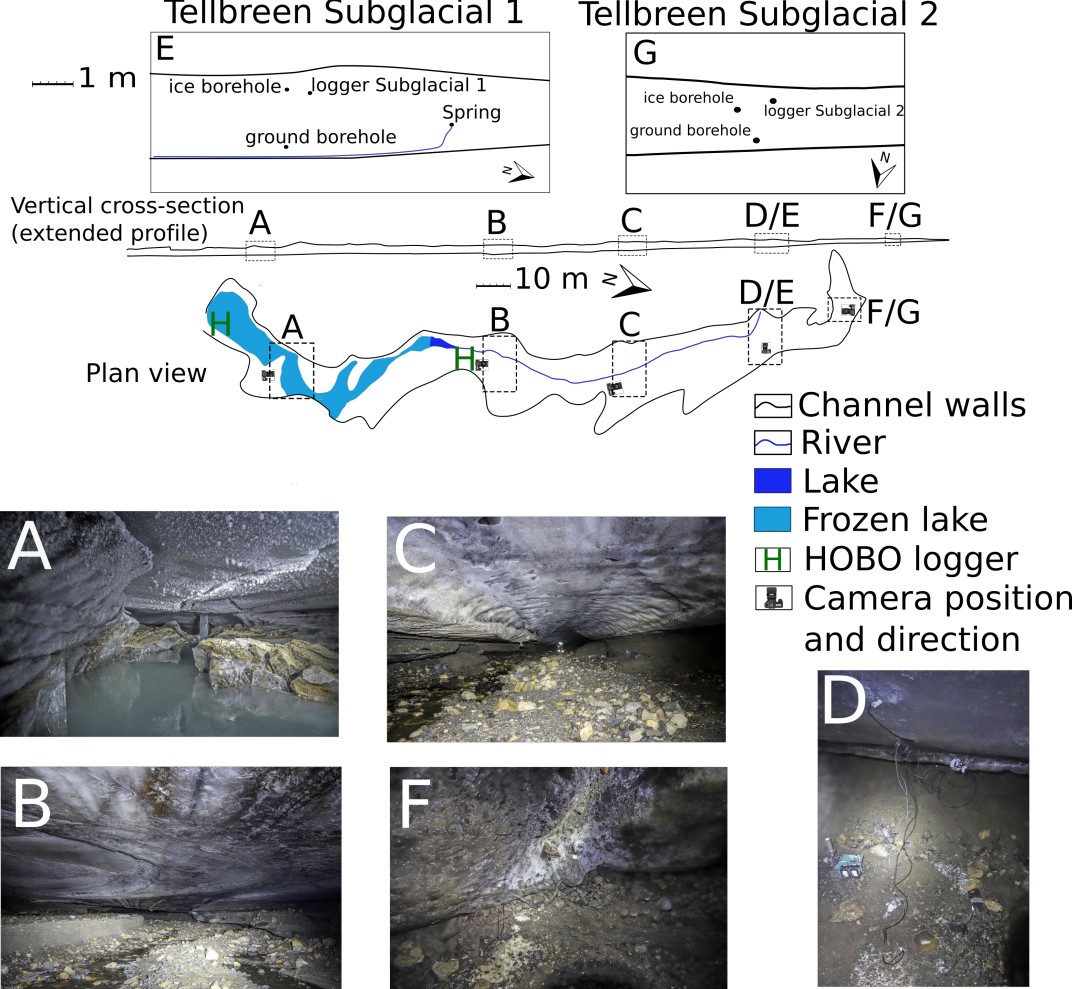

**Figure 3.** Drainage system map of the Tellbreen subglacial drainage system with plan view and picture locations (lower sketch) and vertical cross-section/ extended profile (middle sketch). The channel entrance lies at the south-west, leading into a frozen lake area (A). One Hobo logger (H) was installed next to the entrance. A small stream is running through a Nye-Channel, filling up the lake throughout winter (B). Nye-channel further into the subglacial drainage system with stream running on the left side and a person for scale in the picture centre, at the location of the spring (C). Logger setup next to the spring (D) and at the channel end (E). All pictures taken on 14.03.2019.

2019, thus allowing direct access to it. In the entrance was a partly frozen lake, with an average water depth between 1-2 m. Upstream of the lake, a stream was running through the subglacial channel, incised into the sediment (Nye-channel). A small spring at the end of the channel (see map in figure 3) could be identified as the source of the water with groundwater emerging out of the mountain as possible origin.



## 2.2 Logger installations

Two Tinytag Plus 2 temperature data loggers were installed in the Larsbreen subglacial drainage system on the 21st of May 2016 (approximately 15 m below the ice surface). Both loggers were set to record hourly temperature values. The first data logger had an internally mounted 10K NTC Thermistor sensor (0.01 °C resolution and 0.5 °C accuracy at 0 °C temperature) and was mounted 2 m above the cave floor at the sidewall of the channel (location C in figure 2). The second logger had an external 10K NTC Thermistor sensor (0.02 °C resolution and 0.35 °C accuracy at 0 °C temperature). A hole was drilled into the side of the channel bottom using an electrical drill engine and a sensor was placed at the bottom of the borehole at 1.05 m depth. The hole was then closed with sediment. Data from the data loggers were retrieved in November 2016 before the snow closed the system off again, thus making data recovery very difficult to impossible.

Two different logger systems were used during the Tellbreen installations in 2019. The first system consisted of two small HOBO Pendant Temperature/ Light Data loggers (0.04 °C resolution, ±0.05 °C accuracy). They were set to a 15 min logging interval and mounted to the roof of the channel at the locations marked with an H at the map in figure 3. The second logger system was DK650 Rugged "visual" data loggers from Driesen + Kern GmbH. The loggers (0.01 °C resolution, ±0.03 °C accuracy at 25°C) recorded with one internal temperature sensor and three external PT-1000 temperature sensors. Two loggers set to a 1 min logging interval were installed in the Tellbreen subglacial drainage system in March 2019. The first logger was installed next to the outlet of a small stream of running water, location D in figure 3 (the location is in the following referred to as 'Subglacial 1'). The logger was recording the air temperature in the channel, approximately 1,5 m above the water surface. A 0.9 m deep hole was drilled into the ice at the glacier/bed interface using a battery drill. A second hole was drilled 1 meter deep into the channel bottom sediment and encapsulated with a plastic pipe. Two temperature sensors were inserted, one at 0.9 m depth and the other one at 0.45 m depth. Both ends were sealed with silicone. The second data logger was installed at the end of the drainage system (location E in figure 3), where no water was present (the location is in the following referred to as 'Subglacial 2'). The setting was thereby similar to the other logger. The data were successfully retrieved in October 2019. The HOBO logger at the entrance of the drainage system was lost during the summer melt season and the 0.9 m depth sediment sensor of logger 'Subglacial 1' was malfunctioning. Sediment temperature from this sensor is therefore not recorded.

## 2.3 Subglacial drainage system mapping

Both subglacial drainage systems were mapped using a laser distance meter modified for speleological purposes, also called DistoX (Heeb, 2014). The device measures distance, compass bearing and inclination along a line transect survey and transfers the readings to a PDA device with installed PocketTopo software (Heeb, 2014). Handheld GPS coordinates (accuracy 5 m) served as starting point in both cases. The compass bearings were corrected for the local magnetic declination and the measurements were transformed into UTM coordinates and plotted using the cave mapping software Speleoliti (Dular, 2006).



## 2.4 Meteorological data

Meteorological data from the AROME-Arctic model was used for this work. The AROME-Arctic model is a regional short-range high-resolution forecasting system for the European Arctic with a 2.5 km grid resolution developed by the Norwegian Meteorological Institute (Müller et al., 2017; Køltzow et al., 2019). Forecasted surface variables (e.g. 2 m temperature, 2 m humidity) are interpolated over the grid based on optimal interpolation (Giard and Bazile, 2000). We tested the accordance of the forecasted weather with the actual observed weather for the Svalbard airport for the observation periods in 2016 and 2019. As the meteorological observations from the station at the airport (located at 28 m a.s.l.) showed good agreement with the closest grid point of the AROME-Arctic model (located at 65 m a.s.l.) in the general trends of both air temperature (Supplement figure S1 and table S1) and rainfall (Supplement figure S2 and table S1), we decided to use the weather forecast from the AROME-Arctic model for our study. The closest grid cell from the model to each field site was chosen and data downloaded on an hourly base. The data was subsequently used to calculate daily mean temperatures for the observation periods. The rainfall rates were summed for each day to provide daily amounts of rainfall in mm.

## 2.5 Data processing

All data were processed using MATLAB R2018b. The recorded subglacial temperatures were aggregated from hourly or minute resolution to daily resolution by calculating the daily mean temperatures for each sensor reading. We then used these temperatures to calculate Pearson correlation coefficients for each sensor reading with both the daily mean surface air temperature at 2 m elevation as well as the daily rainfall amount as provided by the AROME-Arctic forecast model. The correlation coefficients were calculated for the whole observation period, as well as on a monthly and seasonal base. The freezing season during the observation period was thereby defined as from the start of the dataset until the onset of persistent above-freezing air temperatures. The latter marks the start of the melting season that lasts until the first days of consecutive air temperatures below freezing. For Larsbreen the freezing season lasted from 22.05.2016 to 19.06.2016 and the melt season from 20.06.2016 to 22.08.2016. The Tellbreen freezing season lasted from 21.03.2019 to 29.06.2019 and the melt season from 30.06.2019 to 15.09.2019.

## 3 Results

We present in the following the results from our measurements. Modelled surface air temperature from the AROME-Arctic model is thereby referred to as "surface air temperature" and the modelled rainfall as "rainfall". The measured air temperatures in the two subglacial channels are referred to as "channel air temperature", the temperatures from the ground boreholes in the subglacial sediments as "sediment temperature" and the temperatures from the boreholes in the ice walls of the channels as



**Figure 4.** Daily subglacial temperatures compared to weather conditions at Larsbreen in 2016. a) Time series of subglacial channel air temperature, modelled surface air temperatures and modelled rainfall. b) Time series of subglacial sediment temperature, modelled surface air temperature and modelled rainfall.

"ice temperature".

## 3.1 Larsbreen

Channel air temperature in the subglacial drainage system at the frontal part of Larsbreen is generally colder than in Tellbreen
5 (following section) and follows a seasonal trend (figure 4). In wintertime, temperature falls below -2.5°C. During the summer melt season channel air temperature remains slightly below 0°C. The channel air temperature starts increasing with the onset of the melt season in late May/early June and quickly increases to close to 0°C, when water starts flowing through the drainage system. Channel air temperature then remains more or less stable until the end of the main melt season in mid-August. From





then on, channel air temperature is fluctuating between -1°C and 0°C, thereby following the trends of the surface air temperature as well as surface rainfall.

The sediment temperature at 1.05 m depth begins increasing with a slight offset compared to the channel air temperature in
the drainage system. The initial increase is slow with about 1°C total warming over 1.5 months and concludes with a sudden increase of 1.4°C to slightly above 0°C between the 18th and the 22nd of July 2016. Temperature drops below 0°C after two weeks, with the sediment cooling back down to almost -1°C until mid-September. This development is followed by several warming events between mid-September and mid-November, when the sediment temperature reaches up to 0°C two more times, followed by subsequent cooling. At the end of November, temperature starts decreasing, thereby following the general
seasonal trend imposed by the surface air temperature.

The comparison with the weather data (see table 1) shows that the channel air temperature is correlated to the surface air temperature (r=0.57 for the whole observation period). During the late freezing season and during the peak melt season, the correlation is weak (r=0.43 in May and r=-0.33 in July). Very strong correlations exist, however, in June and during autumn
(August-November) when the channel air temperature is following closely the surface air temperature. Sediment temperature is very highly correlated to the surface air temperature in May, June and August. Rain has the highest correlation to channel air and sediment temperatures in May and in June. The plot of the signal difference compared to the previous day (i.e. day-to-day difference, Supplement figure S3) shows in addition that the highest temperature change in 1.05 m depth occurs shortly after a rain event on the surface. Thereafter the sediment temperature follows the surface air temperature.

During data recovery in November 2016, it was discovered that the sensor drilled down to 1.05 m into the sediment was eroded during summertime and by then only measuring at around 4 cm depth. Additionally, the whole cave system around it had changed considerably with both a widening and a deepening of the channel, and an additional new meander arm, which had developed at the lateral side of the sensor, allowing the water to pass at larger distance and lower depth compared to the
sensor location.

### 3.2   Tellbreen

The channel air temperature in the middle of the subglacial drainage system at Tellbreen ('Hobo' logger) shows very little variations during late winter and spring (figures 5). Temperature remains more or less constant until July when it suddenly
increases and reaches above 0°C. This increase is then followed by larger variations with five more periods when the temperature in the subglacial drainage system raises above 0°C. The comparison with the meteorological data shows that the first peak of the subglacial channel air temperature is preceded first by heavy rainfall and then by the highest surface air temperature recorded during the melt season. Before this event the channel air temperature of the subglacial drainage system is only weakly correlated with the surface air temperature. After the heavy rainfall and the highest surface air temperature, the channel air







**Figure 5.** Observed subglacial daily mean temperatures at Tellbreen during 2019. Subfigure a) CTime series of subglacial channel air temperature at 'Hobo' location compared to modelled surface air temperature and modelled rainfall. b) Channel air, 0.45 m depth sediment and 0.9 m depth ice temperature at location 'Subglacial 1' next to the source of the running water. c) Channel air, 0.45 m depth sediment, 0.9 m depth sediment and 0.9 m depth ice temperature at location 'Subglacial 2' at the end of the drainage system.

temperature shows, however, a very strong correlation, indicating, that the system is from that point in time on coupled to the atmosphere. The channel air temperature also shows a moderate positive correlation to rainfall during June/ July preceding the coupling of the subglacial system to the atmosphere from mid-July.

5     The channel air and 0.45 m depth sediment temperatures at the logger 'Subglacial 1' (figures 5 and 6), next to the outlet of the small spring, remain generally between -0.2°C and -0.1°C during wintertime. Ice temperature in 0.9 m depth is slightly lower between -0.28°C and -0.24°C during the same time period. The channel air temperature starts increasing in the beginning of July, and from then onwards the channel air temperature of the 'Subglacial 1' logger follows the same trend as the channel



**Figure 6.** Subglacial temperatures at location 'Subglacial 1'. a) Time series of daily mean channel air temperature, modelled surface air temperature and modelled rainfall. b) Time series of daily mean 0.45 m depth sediment temperature, modelled surface air temperature and modelled rainfall. c) Time series of daily mean 0.9 m depth ice temperature, modelled surface air temperature and modelled rainfall.

air temperature at the 'Hobo' logger. The sediment temperature at 0.45 m depth follows this trend with a slight offset, how-ever, showing a sudden and steep temperature decrease during the end of August, before it reaches values similar to channel air temperature again in October. Compared with surface air temperature, the sediment and the channel air temperatures show only weak positive correlation during wintertime, whereas the ice temperature is strongly negative correlated during wintertime

5 (table 1). During June and July the channel air temperature is strongly correlated to the surface rainfall and following August a strong correlation exists between surface air temperature and channel air temperature. The sediment temperature at 0.45 m depth is only moderately correlated to the surface air temperature in July, shows, however, a strong negative correlation to the surface rainfall in July and August, followed by a strong negative correlation to the surface air temperature in October. Ice tem-perature is only weakly correlated to the surface air temperature for most of the observation period, except for an almost perfect



**Figure 7.** Subglacial temperatures at location 'Subglacial 2'. a) Time series of daily mean channel air temperature, modelled surface air temperature and modelled rainfall. b) Time series of daily mean 0.45 m depth sediment temperature, modelled surface air temperature and modelled rainfall. c) Time series of daily mean 0.9 m depth sediment temperature, modelled surface air temperature and modelled rainfall. d) Time series of daily mean 0.9 m depth ice temperature, modelled surface air temperature and modelled rainfall.

negative correlation in October. A moderate negative correlation to surface rainfall exists in July and August. The recovery of the sensor data showed that the sediment sensors were moved during summertime. The whole sediment was displaced about one meter downstream, with the sensors still being embedded in the sediment in October. This might either have been due to a re-deposition of the sediments with the sensors inside or to the sensors floating in a mixture of water and sediment and then

5  being re-buried later on. Re-deposition of the channel bottom sediments was also observed at several other spots within the subglacial drainage system.





At the location of the 'Subglacial 2' data logger (figures 5 and 7), where no running water was present during the visits, the sediment temperatures were higher than the channel air temperature. Sediment temperatures are mostly stable throughout winter, ranging from -0.2°C to -0.1°C. Ice temperature at 0.9 m depth is lower and ranges between -0.4°C and -0.3°C. All observations show a seasonal variation with temperatures rising in summertime. The channel air temperature fluctuates between

-0.3° and -0.2°C in winter, starts increasing end of June, and then remains between -0.2°C and -0.1°C until mid-September. Three major temperature peaks occur at 0.45 m depth in the sediment throughout the melt season that seem to correspond with rain and warm surface events. The Pearson Correlation coefficients (table 1) confirm a very strong positive correlation between surface air temperature and sediment temperature at 0.45 m depth in August and a strong positive correlation between surface rain and sediment temperature at 0.45 m depth in June. The sediment temperature at 0.9 m depth follows a similar, but slightly

weaker correlation. Both sediment temperature series reach peak values at the end of August while the biggest rainfall event of the melt season is recorded on the surface. After this event, both sensors show a strong positive correlation to the surface air temperature (table 1). Both sensors were additionally found to be eroded out of the sediment and hanging in the air during data recovery in October 2019. Ice temperature follows mostly a strong negative correlation with the surface air temperature and has a moderate correlation with rainfall in June (table 1).

## 4 Discussion

Our results suggest that a subglacial drainage systems has two main thermal phases (figure 9), resulting in a seasonal temperature cycle:

During wintertime the drainage system is decoupled from the atmosphere due to ice creep closure and snow infilling of

crevasses and channels closing off the entrances and blocking air and water flow from the surface, which would else allow heat exchange with the atmosphere. During this phase the thermal regime of the subglacial drainage system is controlled by the ice and ground temperatures and the heat flux therein.

The second phase starts with the onset of the melt season and the melting of snow in crevasses and channels, allowing heat

exchange to the subglacial drainage system from the atmosphere. The drainage system is then coupled to the atmosphere and the channel air temperature follows the variations in the atmosphere, similar to observations inside glacier cave systems of the active volcano Mount Hood (Pflitsch et al., 2017). Subsequently, also sediment and ice temperatures at the walls of the drainage system will follow the atmospheric variations through coupling via the channel air temperature. Surface rainfall can additionally influence the heat balance of the drainage system by eroding snow in crevasses and channels and by providing ad-

ditional heat advection. The fact that the thermal regime of the subglacial drainage system and its surrounding borders are now coupled to the surface air temperature also leads to short-lived surface events playing a major role for the thermal regime of the subglacial drainage system. The opening of the subglacial drainage system and the resulting coupling to outside conditions lead to the seasonal warming. Short-lived surface events, however, have a major impact by being able to move the subglacial

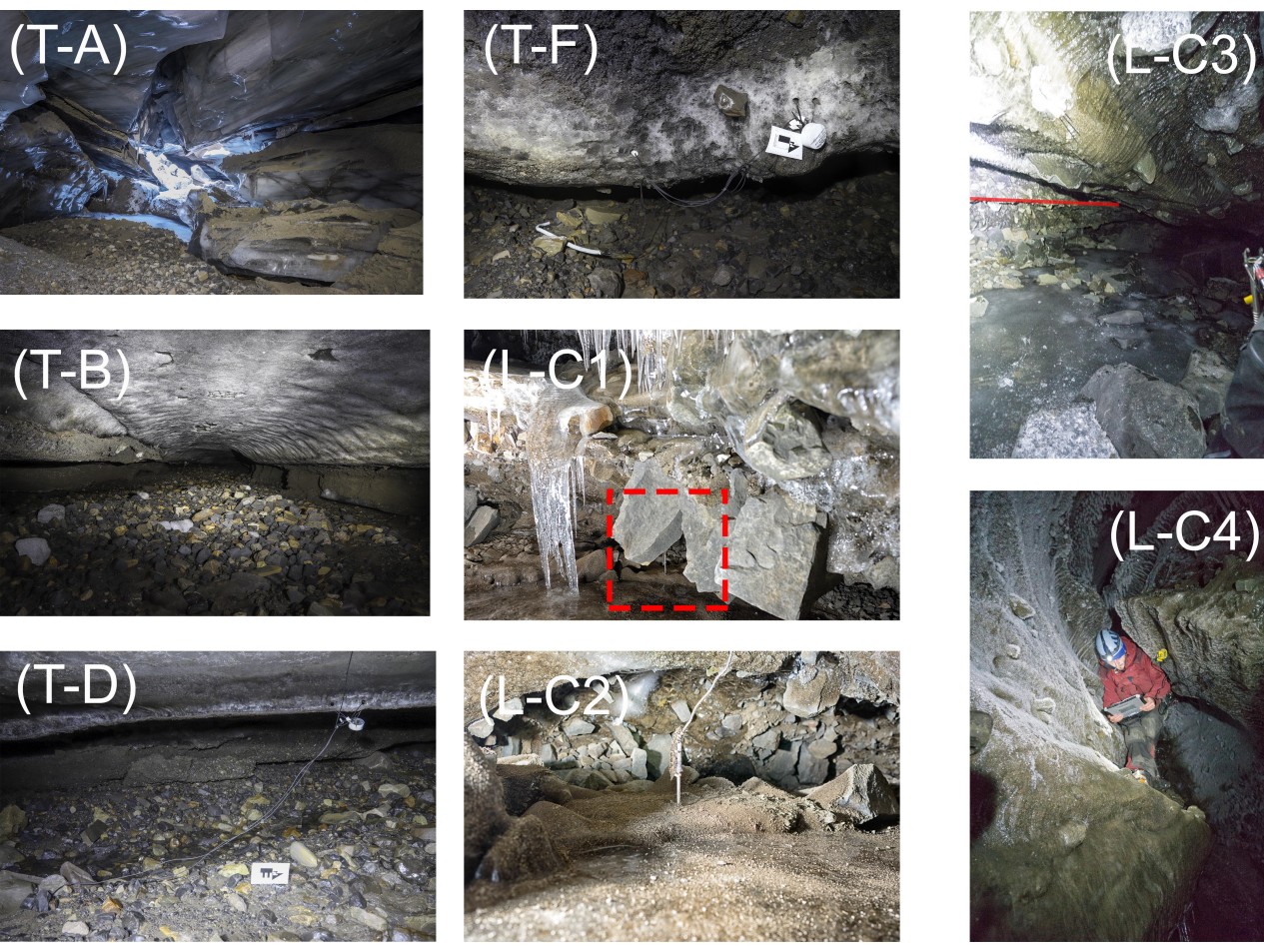

**Figure 8.** Pictures of the Tellbreen and the Larsbreen subglacial drainage system after the summer melt seasons. T-A) Area A from figure 3 at the Tellbreen subglacial drainage system with collapsed roof area. Date: 06.10.2019. T-B) Area B from figure 3 at the Tellbreen subglacial drainage system with stranded icebergs on the higher banks, indicating a flood event. Date 06.10.2019. T-D) Area D from figure 3 with logger 'Subglacial 1' and sediment sensors, which have been moved. Date: 06.10.2019. T-F) Area F from figure 3 with logger 'Subglacial 2', where both sediment loggers were eroded completely out of the sediment. Date: 06.10.2019. L-C1) Area C from figure 2 ??? at the Larsbreen subglacial drainage system. Note the sensor being eroded out of the ground in the lower part of the picture (red frame). Date: 30.11.2016. L-C2) Close-up of the eroded 1.05 m depth sediment sensor at the Larsbreen subglacial drainage system. Date: 30.11.2016. L-C3) Looking downstream at location C from figure 2 inside the Larsbreen subglacial drainage system. Note the channel floor from the previous winter marked in red. Date: 30.11.2016. L-C4) Downloading data from the loggers at location C from figure 2, where the channel has further incised during the melt season. Date: 30.11.2016.



**Figure 9.** Proposed thermal phases of subglacial drainage systems. The y-axis shows the difference between summer- and wintertime, with subglacial temperatures being controlled by surrounding ice and sediment in wintertime and coupled to surface weather during summertime. Presence of running liquid water causes stable temperatures due to forced convection both in summer and in winter. Yellow arrows mark air circulation.

drainage system temporarily out of its balance.





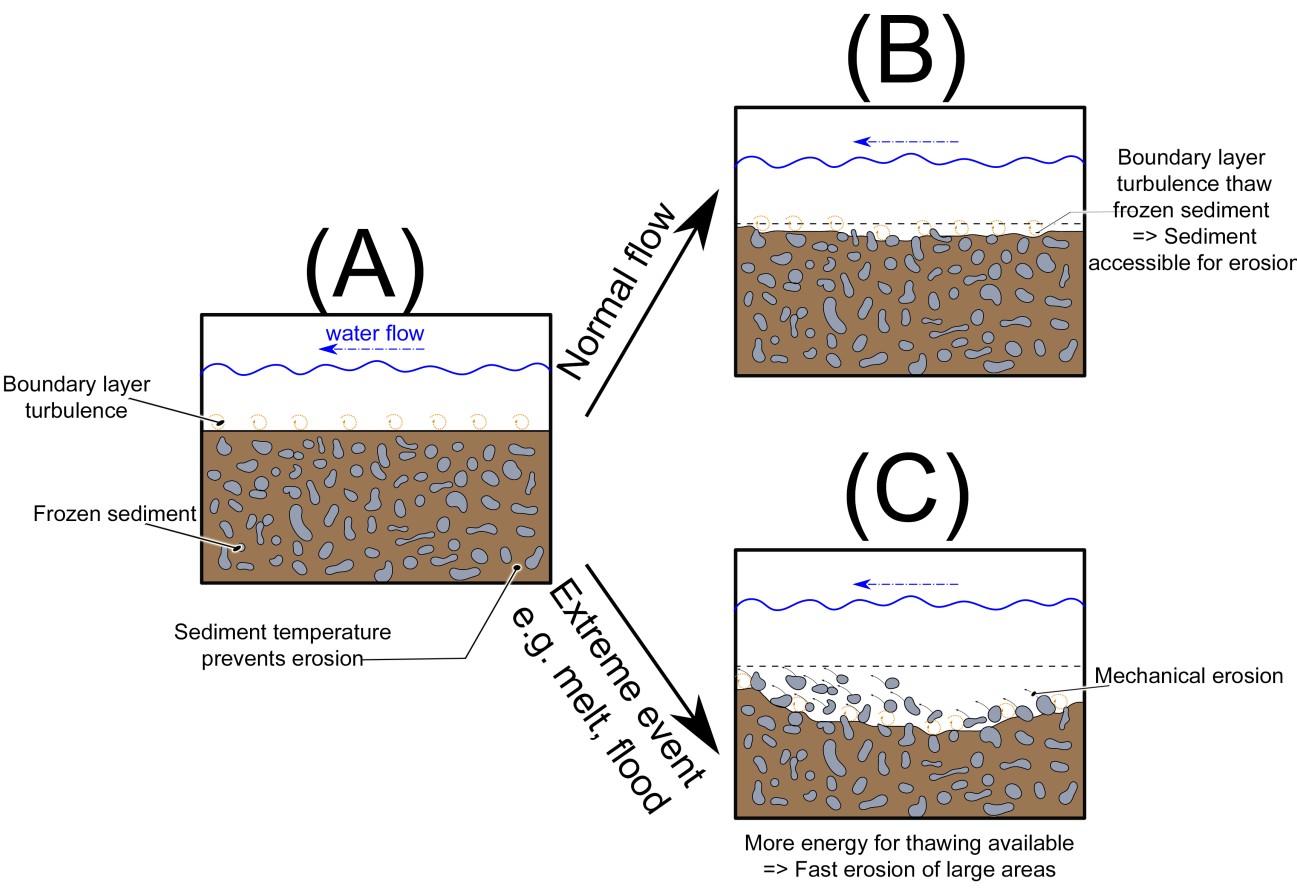

**Figure 10.** Proposed thermo-mechanical erosion mechanism at the bottom of subglacial channels of cold-based glaciers. A: Temperatures below freezing prevent the water to erode material in case of low flow. B: During summer and normal flow rates boundary layer turbulence thaw the surface layer of the sediment and exposes it to mechanical erosion. C: Exceptional high flow velocities and available thermal energy cause a rapid sediment thaw, exposing larger areas of the ground to mechanical erosion and clasts get transported away due to high flow velocities, leading to high erosion rates.

A different case at both phases is the presence of running water. This can especially be seen for the case of the Tellbreen subglacial drainage system. The channel air temperatures at the loggers 'Subglacial 1' and 'Hobo' are stable during wintertime,





whereas the channel air temperature at logger 'Subglacial 2' at the end of the channel is fluctuating throughout the winter. Both loggers 'Subglacial 1' and 'Hobo' were situated next to the small stream of running water, which was present in the subglacial drainage system in wintertime. Logger 'Subglacial 2', however, was situated at the end of the drainage system, further away from the running water. Starting in mid-August, a stabilization of channel air temperature can also be observed

at logger 'Subglacial 2'. The inspection of the subglacial drainage system in October 2019 showed that a new passage had opened behind logger 'Subglacial 2' and water was likely flowing past the logger following mid-August. Similar behavior was observed in the Larsbreen dataset. The air temperature in the subglacial drainage system is fluctuating more until early July. From then onwards it is stable until end of August, before it is fluctuating again. This can be explained by running water in the system from early July until the end of August. Our measurements show that all subglacial drainage systems with water flow

present have stable channel air temperatures, whereas all without water flow have fluctuating air temperatures. This is likely caused by convection mechanisms. In the absence of water flow, airflow inside the drainage system is driven by air temperature differences between different drainage system parts as well as the surface and the subsurface, thus being highly susceptible to temperature perturbations. Water flow on the other hand results in a forced convection, creating a stable airflow and thus heat exchange in the subglacial drainage system, leading to a stable equilibrium temperature (figure 9).

The channel air temperature record from the 'Hobo' logger during summertime seems, at a first glance, to not follow this relationship. Even though water is flowing past the system, creating a forced convection, the channel air temperature is fluctuating more following July. The revisit of the drainage system in October showed that the roof of the area, where the lake had formed during wintertime, had collapsed at some point during the melt season (see figure 8T-A). Based on the peak of the

'Hobo' temperature record, this event seems to haven taken place in mid-July, following several days of rain and the highest daily surface air temperature of the season. Once the roof collapsed, a new and big connection to the surface opened closer to the logger and thus could have led to a change of the circulation system inside the subglacial drainage system.

Based on the observation that the sediment sensors were eroded out in both subglacial drainage systems monitored, we pro-

pose a thermo-mechanical erosion mechanism for subglacial erosion at soft-bedded cold-based glaciers (figure 10), such as also suggested for rivers in permafrost environments in Siberia (e.g., Costard et al., 2003; Randriamazaoro et al., 2007; Dupeyrat et al., 2011; Costard et al., 2014). Sediment is thereby frozen and not exposed to direct mechanical erosion by the water. Turbulent heat exchange at the boundary layer between the subglacial water flow and the bed lead, however, to a warming of the frozen material. Once sufficiently thawed, the stability of the material decreases and fast mechanical erosion by flowing water

can occur. As a result more, previously frozen, material is exposed to the water flow, leading to a thickening of the active layer and higher sediment availability for erosion.

Our observations suggest, that about one meter of the subglacial material was eroded within four days (18.06-22.06.2016) at Larsbreen, following a combination of surface rain and the peak surface air temperature of the season. At Tellbreen, at least 0.9

m erosion occurred on the 30th of August 2019 within one day, caused by a combination of peak surface air temperature and





record rainfall. This event lead to a flood (indicated by stranded ice pieces on higher banks, see picture T-B in figure 8) inside the subglacial drainage system, which had enough erosional power to open new passages and cause particularly high erosion rates. This flood event also caused the erosion and subsequent displacement of the sediment sensors of the 'Subglacial 1' logger. The refreezing of the sediment around the sensor might explain the sudden drop in sediment temperature on the date of the

flood event. Another likely explanation for the sudden sediment temperature drop at the 'Subglacial 1' logger might be a technical problem due to strain on the sensor cable, which also caused the failure of the 0.9 m depth sediment sensor on the same date.

Comparing the change of both subglacial drainage systems over the melt season indicates much higher total erosion of several meters inside the Larsbreen drainage system, whereas the Tellbreen drainage system showed only little visual change. This

coincides with the Tellbreen drainage system being almost horizontal, whereas the Larsbreen drainage system is inclined (see extended profiles in figure 2 and figure 3) leading to higher potential stream power. Higher inclination leads therefore for our two study sites to higher erosion rates, an effect also noted for supraglacial systems (Germain and Moorman, 2019).

Our results highlight the impact of short-lived hydro-meteorological surface events on both subglacial thermal regimes and

subglacial erosion. Increasing summer surface air temperature, as well as increasing frequency and intensity of rainfall events, as they are expected under a changing climate on Svalbard (Hanssen-Bauer et al., 2019) might therefore have strong impacts on cold-based glaciers and adjacent ecosystems. Extreme events in surface air temperature and rainfall can be projected to cause large and rapid erosion events of subglacial material, subsequently leading to peak sediment discharges into proglacial and fjord ecosystems. Furthermore, the higher surface air temperature in the future climatic scenario will also lead to a faster degradation

of subglacial permafrost via vertical active layer expansion due to thermo-mechanical erosion and a general change of glacier thermal regimes in the vicinity of subglacial drainage systems. This can in consequence be of importance for switching small parts of a glacier bed from frozen to unfrozen, potentially leading to a reduction in friction of the glacier bed and thus to glacier instability (e.g., Nuth et al., 2019; Thøgersen et al., 2019). For inefficient drainage systems, where water can spread widely under the glacier, our observations suggest, that meteorological extremes at the glacier surface might have a thermal

impact on large parts of the glacier bed. Such thermal impacts could operate not only by the transition from frozen to unfrozen bed conditions, but also from hard-frozen to plastic-frozen bed conditions, thus enabling increased sediment deformation and glacier motion. Our data also shows that changes in subglacial thermal regime, induced by one extreme event, can occur down to at least one meter subglacial sediment depth, which could cause loss of friction when subglacial material transits from hard-frozen to plastic-frozen or from frozen to unfrozen. This thermally impacted depth would be a sufficient depth to cause basal

motion and thus glacier movement due to sediment deformation as previous studies have shown (e.g., Boulton and Hindmarsh, 1987; Engelhardt et al., 1990; Blake, 1992; Humphrey et al., 1993; Iverson et al., 1994; Boulton, 1996; Engelhardt and Kamb, 1998; Boulton et al., 2001). On overall, our observations therefore suggest that under a changing climate in Svalbard we might see, in specific under increasing hydro-meteorological extremes, increased sedimentation rates downstream of glaciers, as well as increased glacier motion for entirely or partly cold-based glaciers.

## 5   Conclusions

In this study we have measured subglacial permafrost and active layer temperatures in sediment at the base of the subglacial drainage systems of two cold-based glaciers in Svalbard during late winter, spring and the summer melt season. Our investigations reveal colder ground temperatures under Larsbreen and warmer ground conditions under Tellbreen due to the presence of
water flow all year round. Our work highlights the importance of short-lived hydro-meteorological events, in specific melt and rainfall peaks, for the subglacial thermal regime and erosion. This may have further implications for the dynamics of cold-based glaciers. Our two study glaciers suggest those subglacial drainage systems generally undergo a seasonal hydro-thermal cycle with a stagnant phase during wintertime, where the thermal regimes of the subglacial drainage systems and their surroundings are decoupled from the atmosphere and temperatures are dictated by the ice surrounding the channels and the upwards geother-
mal heat flux from the ground. During the summertime the subglacial drainage systems become coupled to the atmosphere with the opening of the channels and subsequent convection associated with air and water flow. Forced convection induced by flowing water leads thereby to stable temperatures, whereas convection due to airflow alone leads to more fluctuating temperatures.

Peaks in surface air temperature and rainfall events had a major impact on the thermal regimes of the glacier beds investigated, and caused rapid and large erosion rates through thermo-mechanical erosion. With summer temperature and rainfall increasing under climatic changes, this process might lead to increased peak sediment input into proglacial and fjord ecosystems. Peaks in summer air temperature and rainfall could well enhance degradation of subglacial permafrost and might thereby potentially also enhance sediment deformation and sliding, through thawing of previously frozen subglacial sediment as well as
transition from hard-frozen to plastic-frozen subglacial sediment. In perspective, our results suggest that the subglacial thermal regime of cold-based glaciers, and related erosional and glacier-dynamical processes may be able to change faster than often anticipated, in particular through direct impact by hydro-meteorological surface conditions and extremes. This calls for more research into the topic, and consequences for glacier evolution and downstream effects.

*Data availability.*   All data will be made available online at the end of the review process.

*Author contributions.*   AA planned the study, conducted the fieldwork, analysed the data and wrote the manuscript. JO helped with the fieldwork at Tellbreen and created the map of Tellbreen. All authors contributed to the interpretation of the data and the manuscript.

*Competing interests.*   The authors declare no competing interests.





*Acknowledgements.* This study was funded by the Research Council of Norway with an Arctic Field Grant in 2019 (Measuring subglacial processes, RiS ID 11156), the Svalbard Environmental Protection fund (project 17/31, Temperature monitoring of Larsbreen ice caves) and by ESA under Permafrost CCI (contract 4000123681/18/I-NB). This study is a contribution to the Svalbard Integrated Arctic Earth Observing System (SIOS). The University Centre in Svalbard provided logistical support and special thanks goes to Sara Cohen and Dag Furberg Fjeld.

5   Andreas Alexander acknowledges field assistance of Simon Polster, Daniel Bader, Paul Petersik, Martha Helle Dybo, Caroline Lasson, Magdalena Korzeniowska, Sandy Curth, Franz Czech, Caroline Hofbauer, Eicke Hecht, Benedikt Ehrenfels, Sander Verbiest, Sebastian Manz, Carsten Hinzmann, Florian Denk, Martin Knoche, Kristian Reed, Arnau Busom Vidal, Martin Thyme Christensen, Alicja Borsberry-Woods and Moritz Oberrauch during the work at Larsbreen. The DistoX used to map the cave system at Larsbreen was provided by Kiya Riverman. Pierre-Marie Lefeuvre and Laurence Dyke assisted during the fieldwork in autumn 2019 at Tellbreen. We further on acknowledge

10  the assistance of Irene Kastner and her amazing dogs Bertha, Runar, Fenris and Aaron during the work at Tellbreen in autumn 2019.



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





The Cryosphere Discussions — Open Access EGU

**Table 1.** Pearson correlation coefficients for all sensor records with surface air temperature and rainfall for the whole time period and sub-periods. Correlation coefficients can be classified after Evans (1996): Very weak (0.00-0.19), weak (0.20-0.39), moderate (0.40-0.59), strong (0.60-0.79) and very strong (0.80-1.0).

| Location | Meteorological data | Sensor reading | Whole period | Freezing period | Melt period | March | April | May | June | July | Aug. | Sept. | Oct. | Nov. |
|---|---|---|---|---|---|---|---|---|---|---|---|---|---|---|
| Larsbreen | Temp. | Channel air | 0.57 | 0.35 | 0.35 | | | 0.43 | 0.9 | −0.33 | 0.87 | 0.69 | 0.76 | 0.66 |
| | | 1.05 m sediment | 0.12 | 0.22 | −0.04 | | | 0.71 | 0.8 | −0.22 | 0.72 | 0.50 | 0.43 | 0.63 |
| | Rain | Channel air | 0.21 | −0.07 | 0.09 | | | 0.54 | 0.54 | 0.13 | 0.26 | 0.48 | 0.42 | 0.24 |
| | | 1.05 m sediment | 0.19 | −0.16 | −0.13 | | | 0.42 | 0.42 | −0.30 | −0.04 | 0.35 | 0.29 | 0.39 |
| Tellbreen Hobo | Temp. | Channel air | 0.62 | 0.44 | 0.74 | −0.26 | 0.43 | 0.07 | 0.27 | 0.37 | 0.82 | 0.83 | 0.55 | |
| | Rain | Channel air | 0.21 | 0.12 | 0.05 | −0.42 | −0.1 | −0.24 | 0.53 | −0.55 | 0.26 | 0.31 | 0.08 | |
| Tellbreen Subglacial 1 | Temp. | Channel air | 0.71 | 0.27 | 0.53 | −0.10 | −0.63 | 0.24 | 0.42 | 0.30 | 0.76 | 0.86 | 0.78 | |
| | | 0.45 m sediment | −0.03 | −0.32 | 0.43 | 0.41 | −0.70 | 0.52 | 0.05 | 0.57 | −0.13 | −0.43 | −0.86 | |
| | | 0.9 m ice | 0.45 | −0.66 | −0.28 | 0.5 | −0.58 | −0.79 | −0.27 | 0.14 | −0.35 | 0.26 | −0.95 | |
| | Rain | Channel air | 0.24 | 0.10 | −0.02 | 0.02 | −0.26 | 0.32 | 0.61 | −0.73 | 0.31 | 0.34 | 0.14 | |
| | | 0.45 m sediment | −0.28 | −0.02 | −0.24 | 0.36 | −0.15 | −0.13 | 0.46 | −0.63 | −0.70 | 0.04 | 0.17 | |
| | | 0.9 m ice | 0.45 | 0.01 | −0.23 | 0.43 | −0.12 | −0.28 | 0.34 | −0.55 | −0.54 | −0.04 | 0.11 | |
| Tellbreen Subglacial 2 | Temp. | Channel air | 0.63 | −0.15 | 0.06 | 0.28 | 0.52 | −0.30 | 0.45 | −0.06 | 0.68 | 0.79 | 0.22 | |
| | | 0.45 m sediment | 0.65 | −0.02 | 0.59 | 0.41 | 0.40 | −0.62 | 0.42 | −0.004 | 0.85 | 0.77 | 0.77 | |
| | | 0.9 m sediment | 0.34 | −0.46 | 0.28 | 0.39 | 0.62 | −0.74 | 0.06 | 0.03 | 0.60 | 0.79 | 0.73 | |
| | | 0.9 m ice | 0.35 | −0.68 | −0.44 | 0.49 | 0.70 | −0.62 | 0.01 | 0.13 | −0.66 | 0.39 | −0.90 | |
| | Rain | Channel air | 0.19 | 0.12 | −0.18 | 0.43 | 0.005 | 0.05 | 0.54 | −0.50 | −0.16 | 0.30 | 0.16 | |
| | | 0.45 m sediment | 0.23 | 0.21 | 0.03 | 0.40 | 0.15 | −0.22 | 0.60 | −0.29 | 0.16 | 0.19 | 0.11 | |
| | | 0.9 m sediment | 0.31 | 0.13 | 0.32 | 0.48 | −0.07 | −0.32 | 0.59 | −0.22 | 0.46 | 0.22 | 0.10 | |
| | | 0.9 m ice | 0.35 | 0.01 | −0.10 | 0.36 | −0.12 | −0.35 | 0.40 | −0.49 | 0.23 | 0.16 | 0.09 | |