# Peer review of "Subglacial permafrost dynamics and erosion inside subglacial channels driven by surface events in Svalbard"

_The Cryosphere, 2020_

## Short Comment (SC1) · 12 Jul 2020

**Open discussion comments on Alexander et al. (2020)**

This very interesting study provides timeseries of subglacial channel temperatures and erosion under two cold based valley glaciers on Svalbard. It indicates a link between meteorological events at the surface and the yet poorly understood soft bed subglacial processes affecting glacier hydrology and potentially basal slip under cold based ice. The highlighted importance of extreme events is especially relevant in the context of understanding the effects of climatic change on Svalbard.

[Figure]

**1.** The study uses sudden changes in sediment temperature to identify certain erosion events, such as the August $30^{th}$ event for the Tellbreen "subglacial 1" sensor (**p.18 l.35**) and the late July unearthing of the 1.05 m sensor under Larsbreen (**p.18 l.33**). However, other variations in measured sediment temperature are not addressed much in the paper, while it seems they could contribute to the compelling case for the occurrence of episodes of strong erosion linked to surface events. For example, on **Figure 6** the 0.45 sediment temperature follows the channel temperature very closely from the the late June / early July peak rainfall event onward. Could it be possible that this event eroded the channel bed down quite close to the buried sensor, in addition to coupling the subglacial conduits to the atmosphere? Similarly, both 'subglacial 2' sensors in **Figure 7** register a step-wise temperature increase when surface melt starts to occur around June $24^{th}$ and the 0.45 sensor shows more variation after the late June rainfall. Both the 0.45 and 0.9 sensors vary with channel air temperature after the second major rainfall event of August $30^{th}$, and are exposed upon recovery. Maybe it can be argued that these observations point towards distinct episodes of stream erosion occurring over the summer season?

**2. page 19, last paragraph**
In the way I understand the proposed thermo-mechanical erosion mechanism (**Figure 10**), it relies on high stream power to produce high rates of permafrost melting and erosion. The mechanism is especially effective after extreme rainfall and melt events, and applies to the channels of an efficient drainage system, which is where the measurements occurred.
On **lines 23 to 25**, the paper mentions that a more inefficient drainage system would allow more widespread influence of extreme events on basal slip. It would be nice to clarify what is meant exactly, as it seems that in a fully inefficient distributed drainage system, water flow velocities would be too low to allow for turbulent heating and the thermo-mechanical erosion mechanism to occur. It could be interesting to consider Rippin et al. (2005), as they suggest that after mass build-up, pressurized and fast

water flow through the cold based margin sediments could increase local ice velocities. This seems like it could be a situation where the mechanism presented in this study would be quite relevant.

**3.** A final short remark is that in a recent paper, Haga et al. (2020) mention the potential importance of an efficient drainage system in the partial freezing of the Negribreen glacier terminus to its bed surface. The rapid erosion in response to surface events in this study could indicate the capacity of a drainage system to adjust rapidly to changes, even in permafrost. Maybe such an adjustable system is necessary for the cold based conditions of many Svalbard glaciers termini to form, or at least facilitates formation?

**References**

Alexander, A., Obu, J., Schuler, T. V., Kääb, A., and Christiansen, H. H. (2020). "Subglacial permafrost dynamics and erosion inside subglacial channels driven by surface events in Svalbard". In: *The Cryosphere Discussions* 2020, pp. 1–26. doi: 10.5194/tc-2020-124. url: https://www.thecryosphere- discuss.net/tc-2020-124/.
Haga, O. N., McNabb, R., Nuth, C., Altena, B., Schellenberger, T., and Kääb, A. (2020). "From high friction zone to frontal collapse: dynamics of an ongoing tidewater glacier surge, Negribreen, Svalbard". In: *Journal of Glaciology*, pp. 1–13. doi: 10.1017/jog.2020.43.
Rippin, D., Willis, I., Arnold, N., Hodson, A., and Brinkhaus, M (2005). "Spatial and temporal variations in surface velocity and basal drag across the tongue of the polythermal glacier midre Lovenbreen, Svalbard". In: *Journal of Glaciology* 51.175, pp. 588–600. doi: 10.3189/172756505781829089.

---

## Referee Comment (RC1) · Douglas Benn (Referee) · 23 Jul 2020

This is a worthwhile paper, that presents novel and useful data on air and ground temperatures within subglacial conduits under a cold-based glacier tongue in Svalbard. The data include good cave maps and very useful temperature series from a number of sites, spanning both warm and cold parts of the year. The paper is very clearly written and structured, and most is ready for publication without revision. The only shortcomings with the paper concern how it is placed in the context of previous work, and the significance of the some of the conclusions, which is rather over-stated in the closing paragraph of the Discussion.

Detailed comments:

Page 2, Line 26: high sedimentation rates in polythermal and cold-based glacier catchments.

The authors have missed the most important factor concerning the sediment budget of these glaciers: most of the examples cited are either surge-type or were more dynamically active during the Little Ice Age. The Hodgkins study focused on Finsterwalderbreen (surge-type); in Hallet's global compilation the Svalbard examples are surge-type; Etzelmüller looked at Larsbreen and Longyearbreen, both of which were more dynamically active in the past (see Sevestre et al., 2015 regarding former dynamics of Longyearbreen). Sollid and Sørbel conducted a palaeo-study and inferred the glacier thermal regime, so this does not provide independent evidence of the link between thermal regime and sediment dynamics. My point is that the high sediment load on Svalbard glaciers mostly relates to past surges, in which sediment can be elevated to high levels by thrusting and other processes. This sediment is then released and reworked by fluvial and gravitational processes during quiescence. Papers by Lovell should be cited in this respect. (e.g. 1: Lovell, H., Fleming, E.J., Benn, D.I., Hubbard, B., Lukas, S. and Naegeli, K., 2015. Former dynamic behaviour of a cold-based valley glacier on Svalbard revealed by basal ice and structural glaciology investigations. Journal of Glaciology, 61(226), pp.309-328. 2: Lovell, H., Benn, D.I., Lukas, S., Ottesen, D., Luckman, A., Hardiman, M., Barr, I.D., Boston, C.M. and Sevestre, H., 2018. Multiple Late Holocene surges of a High-Arctic tidewater glacier system in Svalbard. Quaternary Science Reviews, 201, pp.162-185. 3: Lovell, H., Fleming, E.J., Benn, D.I., Hubbard, B., Lukas, S., Rea, B.R., Noormets, R. and Flink, A.E., 2015. Debris entrainment and landform genesis during tidewater glacier surges. Journal of Geophysical Research: Earth Surface, 120(8), pp.1574-1595.).

Lines 28-9: This statement implies that there is a 'missing' process of sediment erosion. This is not the case. Boulton (1972) is a very old source with regard to erosion mechanisms; much more recent and comprehensive sources can be cited, which

Interactive
comment

give more attention to fluvial processes. Additionally, the erosional capability of subglacial channels under cold glaciers (and Tellbreen in particular) was flagged up by Naegeli et al. 2014. Dendritic subglacial drainage systems in cold glaciers formed by cut‐and‐closure processes. Geografiska Annaler: Series A, Physical Geography, 96(4), pp.591-608.

Page 4, line 9: Recent work on Tellbreen should be cited here, to provide proper context for the study. Key facts from the following papers should be summarised in a sentence or two at this point in the paper:

Origin of the subglacial channels in Tellbreen: Naegeli et al. 2014 (cited above), and in other Svalbard glaciers: Gulley, J.D., Benn, D.I., Müller, D. and Luckman, A., 2009. A cut-and-closure origin for englacial conduits in uncrevassed regions of polythermal glaciers. Journal of Glaciology, 55(189), pp.66-80.

Dynamical history of Tellbreen: Lovell, H., Fleming, E.J., Benn, D.I., Hubbard, B., Lukas, S. and Naegeli, K., 2015. Former dynamic behaviour of a cold-based valley glacier on Svalbard revealed by basal ice and structural glaciology investigations. Journal of Glaciology, 61(226), pp.309-328.

Page 14: Discussion

Most of the Discussion is well written, building a set of sound conclusions and inferences from the data. The sections on subglacial channel erosion are especially welcome. This process has been previously inferred from the existence of subglacial channels at Tellbreen (Naegeli et al.), but the present paper adds valuable insights about processes and rates. However, two points in the Discussion need attention:

Page 19, line 3-6. It is difficult to see how refreezing of the sediment should cause such a catastrophic drop in sensor temperature. A phase change from liquid to solid in surrounding saturated sediment should result in a temperature increase, not a drop, because freezing gives up latent heat. This feature of the record almost certainly reflects sensor malfunction.

Page 19, lines 21-34. From this point on, the Discussion loses its grip on reality and wanders off into wild speculation. Beneath thinning Svalbard glaciers, the thermal trend is from warm to cold-based conditions, as diminishing ice thickness allows conductive losses to the surface to increase during winter. The authors have convincingly shown that this trend can be reversed locally by the presence of channels which advect additional heat to the bed from the surface during the summer months. There is nothing in the data that indicate that these highly localised and seasonal changes could impact the broader hydrological system or dynamics. Indeed, Tellbreen, like the majority of small glaciers in Svalbard, has strongly negative surface mass balance and is in terminal decline. The trends of thinning ice and permafrost aggradation will continue regardless of local seasonal heating around surface-fed conduits. The paper does not need vague speculation about wider 'impact' in order to be relevant - indeed, the paper is weakened by it. Just end the Discussion at line 21.

---

## Referee Comment (RC2) · Anonymous Referee #2 · 25 Jul 2020

This manuscript provides new information concerning how temperature variability in subglacial channels can impact fluvial erosion beneath cold-based glaciers in Svalbard. I'm not aware of a similar dataset and the results should be of interest to a broad community of glacier and permafrost researchers. While I think the paper should be published eventually, I'd like to see the authors more closely situate their manuscript within the modern published literature on the sedimentology and hydrology of cold ice glaciers in Svalbard, provide a clearer description of the sensor installations in the methods section, and more clearly link the discussion to their results. More detailed comments are included with page and line numbers below.

[Figure]

Page 1, Lines 15-20 – Fluvial incision of subglacial tills can erode sediment, but vertical incision of subglacial channels can become limited by boulder armoring. Fine grained materials are preferentially winnowed from till channel by flow and larger boulders and rocks accumulate on the floor (See Gulley et al., 2014). Because flow cannot mobilize these sediments, vertical incision largely ceases but the channel can still migrate and incise laterally. In the case of the till beneath cold-based glaciers in Svalbard, much of the sediment being eroded by streams was not produced beneath cold based glaciers, as seems to be implied by the authors, but instead is derived from past temperate basal regime or perhaps surges.

Page 2, Lines 25 – Boulton 1972 was written when it was widely believed that cold-based glaciers lacked active subglacial hydrological systems. Hodgkins (1997) highlighted water flow and erosion beneath cold ice in Svalbard. More recent work has shown that much of the water flowing out from under cold-based glaciers is likely derived from subglacial channels which began life as supraglacial streams and later incised through cold ice to reach glacier beds (see Gulley et al., 2009).

Page 4, Figure 1b and c – drawing an outline of the subglacial channel beneath Tellbreen in one panel and only showing the survey line of the channel beneath Larsbreen in the other panel is confusing. At first glance, the cave beneath Tellbreen looks like it is a giant loop! Also, "profile" is typically used to describe what the authors are calling "vertical cross-section (extended profile)" and cross-section is typically used to describe the cross-section of a passage segment (drawn perpendicular to the passage direction).

Page 4, Line 7 – remove "is reported"

Pages 5,6 - Figures 2 and 3 – there is too much information jammed into these figures. I recommend separating the maps from the photo panels and making the photo panels larger. The pictures are too small to see the information being described in them. For example, there is a logger setup mentioned in Fig 3C – but I can't see it.

Methods – HOBO pendant loggers have an accuracy of 0.53 deg C when used for above-freezing temperatures. Accuracy decreases to 0.75 deg C between 0 and -20C https://www.onsetcomp.com/products/data-loggers/ua-002-64/

Page 7, line 20 and 21 – I don't understand this installation. Was the pipe filled with silicone? If so, please make it clear why. Why were the temperature sensors not buried directly in sediment? For the logger installed into the ice, was it also installed in pvc pipe? What diameter holes were drilled in the ice and sediment and what was the diameter of the pipe? A clear line drawing showing the various types of instrumentation as installed in the caves would be very helpful in visualizing the experiment and this information is critical for interpreting the temperature signals.

Page 7, Line 30 – please describe the calibration procedure used for the DistoX.

Page 9, Fig 4 – It would really help visualize relationships between outside air temperature and the cave/sediment temperatures if they were plotted at the same scale.

Page 14, line 5 – the authors keep referring to "winter" but the dataset only runs from March until October. It would be nice if the timeseries graphs could be truncated at the end of the data collection period instead of having 1/5 of each graph displaying no information.

Page 14, line 19 – considering how thin the ice is here, creep closure is probably negligible in controlling the isolation of the cave atmosphere in winter. Snow plugs in the entrances are far more likely.

Page 16, Figure 9 – I don't understand the conceptual model for air flow. During summer, outside air is warmer than the cave. Air should be cooled in the upper entrance and then flow out of the lower entrance. If the cave were not plugged with snow, the opposite flow would occur in winter. I also don't understand how you have a channel in summer that lacks a stream.

Page 17, Figure 10 – mechanical erosion conjures images of plucking or abrasion of

bedrock by suspended bed load. Perhaps fluvial erosion is a better term?

Page 18, Lines 32-35 – maybe I'm missing something, but I cannot figure out what data the authors used to infer the rapid incision described here. Please clearly link this interpretation to the data.

Page 19, Lines 14-34 – Much of this is extremely improbable. . ...

References Gulley, J.D., Spellman, P.D., Covington, M.D., Martin, J.B., Benn, D.I. and Catania, G., 2014. Large values of hydraulic roughness in subglacial conduits during conduit enlargement: implications for modeling conduit evolution. Earth Surface Processes and Landforms, 39(3), pp.296-310. Gulley, J.D., Benn, D.I., Müller, D. and Luckman, A., 2009. A cut-and-closure origin for englacial conduits in uncrevassed regions of polythermal glaciers. Journal of Glaciology, 55(189), pp.66-80. Humlum, O., Elberling, B., Hormes, A., Fjordheim, K., Hansen, O.H. and Heinemeier, J., 2005. Late-Holocene glacier growth in Svalbard, documented by subglacial relict vegetation and living soil microbes. The Holocene, 15(3), pp.396-407. Larsen, N.K., Piotrowski, J.A., Christoffersen, P. and Menzies, J., 2006. Formation and deformation of basal till during a glacier surge; Elisebreen, Svalbard. Geomorphology, 81(1-2), pp.217-234.

---

## Author Comment (AC1) · 30 Jul 2020

First of all we would like to thank Doug Benn for reviewing our manuscript and provide helpful and constructive feedback, which will certainly help to improve the paper. In the following we present our responses to the referee comments and how we will address these in the revision of the manuscript.

The referee comments are presented in ***bold and italic***, our replies follow immediately thereafter.

[Figure]

**General comments**

*This is a worthwhile paper, that presents novel and useful data on air and ground temperatures within subglacial conduits under a cold-based glacier tongue in Svalbard. The data include good cave maps and very useful temperature series from a number of sites, spanning both warm and cold parts of the year. The paper is very clearly written and structured, and most is ready for publication without revision.*

We thank the reviewer for this overall positive judgment of our manuscript.

*The only shortcomings with the paper concern how it is placed in the context of previous work, and the significance of the some of the conclusions, which is rather over-stated in the closing paragraph of the Discussion.*

We thank the reviewer for pointing out the shortcomings of our manuscript in regards to the literature and will improve the manuscript with the suggestions from Doug Benn, as well as the second anonymous reviewer (see response to RC2). We will further on remove the scrutinized closing paragraph of the discussion part from the manuscript (P19 lines 21-34).

**Detailed comments**

*Page 2, Line 26: high sedimentation rates in polythermal and cold-based glacier catchments. The authors have missed the most important factor concerning the sediment budget of these glaciers: most of the examples cited are either surge-type or were more dynamically active during the Little Ice Age.*

We thank the reviewer for pointing out this factor. We have indeed missed it.

*The Hodgkins study focused on Finsterwalder-breen (surge-type); in Hallet's global compilation the Svalbard examples are surge-type; Etzelmüller looked at Larsbreen and Longyearbreen, both of which were more dynamically active in*

*the past (see Sevestre et al., 2015 regarding former dynamics of Longyearbreen).*

While we see the reviewers point regarding the Hodgkins and Hallet studies, we still consider Etzelmüllers' study of Larsbreen and Longyearbreen as relevant literature, as Larsbreen is one of the two study sites of this manuscript.

*Sollid and Sørbel conducted a palaeo-study and inferred the glacier thermal regime, so this does not provide independent evidence of the link between thermal regime and sediment dynamics. My point is that the high sediment load on Svalbard glaciers mostly relates to past surges, in which sediment can be elevated to high levels by thrusting and other processes. This sediment is then released and reworked by fluvial and gravitational processes during quiescence. Papers by Lovell should be cited in this respect. (e.g. 1: Lovell, H., Fleming, E.J., Benn, D.I., Hubbard,B., Lukas, S. and Naegeli, K., 2015. Former dynamic behaviour of a cold-based valley glacier on Svalbard revealed by basal ice and structural glaciology investigations. Journal of Glaciology, 61(226), pp.309-328. 2: Lovell, H., Benn, D.I., Lukas, S., Otte-sen, D., Luckman, A., Hardiman, M., Barr, I.D., Boston, C.M. and Sevestre, H., 2018.Multiple Late Holocene surges of a High-Arctic tidewater glacier system in Svalbard. Quaternary Science Reviews, 201, pp.162-185. 3: Lovell, H., Fleming, E.J., Benn, D.I.,Hubbard, B., Lukas, S., Rea, B.R., Noormets, R. and Flink, A.E., 2015. Debris entrainment and landform genesis during tidewater glacier surges. Journal of Geophysical Research: Earth Surface, 120(8), pp.1574-1595.).*

*Lines 28-9: This statement implies that there is a 'missing' process of sediment erosion. This is not the case. Boulton (1972) is a very old source with regard to erosion mechanisms; much more recent and comprehensive sources can be cited, which give more attention to fluvial processes. Additionally, the erosional capability of sub-glacial channels under cold glaciers (and Tellbreen in particular) was flagged up by Naegeli et al. 2014. Dendritic subglacial drainage systems in cold glaciers formed bycutâ ÌĘAÊĞRandâ ÌĘAÊĞRclosure processes.*

*Geografiska Annaler: Series A, Physical Geogra-phy, 96(4), pp.591-608.*

We will rework the paragraph in discussion (P2, L25-29), remove the reference to Boulton (1972) and instead refer to the literature and the processes nicely explained by Doug Benn in this helpful comment.

*Page 4, line 9: Recent work on Tellbreen should be cited here, to provide proper context for the study. Key facts from the following papers should be summarised in a sentence or two at this point in the paper: Origin of the subglacial channels in Tellbreen: Naegeli et al. 2014 (cited above), and in other Svalbard glaciers: Gulley, J.D., Benn, D.I., Müller, D. and Luckman, A., 2009.A cut-and-closure origin for englacial conduits in uncrevassed regions of polythermalglaciers. Journal of Glaciology, 55(189), pp.66-80.Dynamical history of Tellbreen: Lovell, H., Fleming, E.J., Benn, D.I., Hubbard, B.,Lukas, S. and Naegeli, K., 2015. Former dynamic behaviour of a cold-based val-ley glacier on Svalbard revealed by basal ice and structural glaciology investigations. Journal of Glaciology, 61(226), pp.309-328.*

We will add a few sentences in the paragraph, describing the study site at Tellbreen, to include information from the two papers mentioned by the referee.

*Most of the Discussion is well written, building a set of sound conclusions and inferences from the data. The sections on subglacial channel erosion are especially welcome. This process has been previously inferred from the existence of subglacial channels at Tellbreen (Naegeli et al.), but the present paper adds valuable insights about processes and rates.*

We thank the reviewer for this positive feedback.

*However, two points in the Discussion need attention: Page 19, line 3-6. It is difficult to see how refreezing of the sediment should cause such a catastrophic drop in sensor temperature. A phase change from liquid to solid in surrounding saturated sediment should result in a temperature increase, not a drop, because*

*freezing gives up latent heat. This feature of the record almost certainly reflects sensor malfunction.*

We thank the reviewer for pointing this out and will remove the sentence regarding refreezing of the sediment as a potential explanation of the temperature drop (P19, L4-5).

*Page 19, lines 21-34. From this point on, the Discussion loses its grip on reality and wanders off into wild speculation. Beneath thinning Svalbard glaciers, the thermal trend is from warm to cold-based conditions, as diminishing ice thickness allows conductive losses to the surface to increase during winter. The authors have convincingly shown that this trend can be reversed locally by the presence of channels which advect additional heat to the bed from the surface during the summer months. There is nothing in the data that indicate that these highly localised and seasonal changes could impact the broader hydrological system or dynamics. Indeed, Tellbreen, like the majority of small glaciers in Svalbard, has strongly negative surface mass balance and is in terminal decline. The trends of thinning ice and permafrost aggradation will continue regardless of local seasonal heating around surface-fed conduits. The paper does not need vague speculation about wider 'impact' in order to be relevant - indeed, the paper is weakened by it. Just end the Discussion at line 21.*

We will remove the scrutinized discussion section from the manuscript (P19, L21-34) and end the discussion at line 21, as nicely suggested by the referee.

---

## Author Comment (AC2) · 30 Jul 2020

We would like to thank the anonymous referee for taking the time to review our manuscript and for providing thorough and helpful feedback, which will certainly help to improve the quality of the manuscript.

In the following we present our responses to the referee comments and how we will address these in the revision of the manuscript.

The referee comments are presented in *bold and italic*, our replies follow immediately thereafter.

**Overall comments**

This manuscript provides new information concerning how temperature variability in subglacial channels can impact fluvial erosion beneath cold-based glaciers in Svalbard. I'm not aware of a similar dataset and the results should be of interest to a broad community of glacier and permafrost researchers.

We thank the referee for this overall positive judgment of our dataset.

While I think the paper should be published eventually, I'd like to see the authors more closely situate their manuscript within the modern published literature on the sedimentology and hydrology of cold ice glaciers in Svalbard, provide a clearer description of the sensor installations in the methods section, and more clearly link the discussion to their results. More detailed comments are included with page and line numbers below.

We thank the referee for his feedback. We will rework the introduction of our manuscript to incorporate more modern literature, as requested both by referee 1 and 2 following their suggestions (see also response to RC1). We will further on clarify the sensor installations in the methods section and improve the discussion part following the suggestions from both referees.

**Detailed comments**

Page 1, Lines 15-20 – Fluvial incision of subglacial tills can erode sediment, but vertical incision of subglacial channels can become limited by boulder armoring. Fine grained materials are preferentially winnowed from till channel by flow and larger boulders and rocks accumulate on the floor (See Gulley et al., 2014). Because flow cannot mobilize these sediments, vertical incision largely ceases but the channel can still migrate and incise laterally. In the case of the till beneath

TCD
cold-based glaciers in Svalbard, much of the sediment being eroded by streams was not produced beneath cold based glaciers, as seems to be implied by the authors, but instead is derived from past temperate basal regime or perhaps surges.

This has also been remarked by referee 1 and we will rework the introduction of the manuscript to indicate the source of the subglacial sediment from past dynamic behavior.

Page 2, Lines 25 – Boulton 1972 was written when it was widely believed that cold-based glaciers lacked active subglacial hydrological systems. Hodgkins (1997) highlighted water flow and erosion beneath cold ice in Svalbard. More recent work has shown that much of the water flowing out from under cold-based glaciers is likely de-rived from subglacial channels which began life as supraglacial streams and later incised through cold ice to reach glacier beds (see Gulley et al., 2009).

We thank the referee for pointing this out. We will rework this paragraph (P2, L24-29) to remove Boulton 1972 and account for modern literature and sediment sources. See also the response to referee 1.

Page 4, Figure 1b and c – drawing an outline of the subglacial channel beneath Tellbreen in one panel and only showing the survey line of the channel beneath Larsbreen in the other panel is confusing. At first glance, the cave beneath Tellbreen looks like it is a giant loop! Also, "profile" is typically used to describe what the authors are calling "vertical cross-section (extended profile)" and cross-section is typically used to describe the cross-section of a passage segment (drawn perpendicular to the passage direction).

The reason for drawing the centerline at Larsbreen and the outline at Tellbreen is more of a technical nature. The Larsbreen cave has been mapped with a DistoX and pen and paper, whereas the Tellbreen cave has been mapped using an additional
PDA, allowing for additional survey shots. The post-processing software Speleoliti has, however, not allowed to export the centerline without the survey shots. This is the reason for the outline of the Tellbreen cave instead of the centerline on the map. We can, however, manually trace the centerline within GIS to produce a similar centerline plot as in the Larsbreen case. The naming conventions used for the cave plots originate from the software Speleoliti. We will change the names, following the suggestions from the reviewer.

**Page 4, Line 7 – remove "is reported"**

We will remove it.

Pages 5,6 - Figures 2 and 3 – there is too much information jammed into these figures. I recommend separating the maps from the photo panels and making the photo panels larger. The pictures are too small to see the information being described in them. For example, there is a logger setup mentioned in Fig 3C – but I can't see it.

We understand well that these figures are at the upper end busy. Reviewer 1 finds them "good", and we prefer thus to not split them up and generate more figures and thus a longer manuscript. However, we will try to make the cave maps clearer. We will also formulate the figure caption clearer, as the mentioned logger setup is indeed not in figure 3C, but in figure 3D.

Methods – HOBO pendant loggers have an accuracy of 0.53 deg C when used forabove-freezing temperatures. Accuracy decreases to 0.75 deg C between 0 and -20Chttps://www.onsetcomp.com/products/data-loggers/ua-002-64/

We thank the referee for pointing this out, as we indeed overlooked it. We will add this to the manuscript.

Page 7, line 20 and 21 – I don't understand this installation. Was the pipe filled
with silicone? If so, please make it clear why. Why were the temperature sensors not buried directly in sediment? For the logger installed into the ice, was it also installed in pvc pipe? What diameter holes were drilled in the ice and sediment and what was the diameter of the pipe? A clear line drawing showing the various types of instrumentation as installed in the caves would be very helpful in visualizing the experiment and this information is critical for interpreting the temperature signals.

Boreholes were drilled with a 20 mm diameter drill. In the case of Tellbreen PVC pipe (20 mm diameter) was used to be able to recover the sensors at the end of the measurement period. The pipes had the further benefit to allow for exact depth placement of the sensors, by fixing them in position inside the pipe. Silicone was only used to seal both ends of the PVC pipe in order to prevent water from draining into the pipe (the pipes remained air filled). Both, sediment and ice sensors, were installed in a PVC pipe. In case of Larsbreen, the sediment sensor was buried directly into sediment. We will add additional information to clarify this in the methods part (P7, L2-25). We will further on provide a line drawing outlining the instrumentation setup.

**Page 7, Line 30 – please describe the calibration procedure used for the DistoX.**

We followed the standard procedures for the DistoX calibration as outlined in the DistoX manual. https://paperless.bheeb.ch/download/CalibrationManual.pdf We will add according text in the manuscript.

**Page 9, Fig 4 – It would really help visualize relationships between outside air temperature and the cave/sediment temperatures if they were plotted at the same scale.**

We produced such a figure, see the below figure (Fig. 1). As the cave temperatures appear at the scale of the outside temperatures almost as a straight line without much information given, we prefer to leave the temperatures in Fig 4 at different scales to allow for easier interpretation of the results.
Page 14, line 5 – the authors keep referring to "winter" but the dataset only runs from March until October. It would be nice if the time series graphs could be truncated at the end of the data collection period instead of having 1/5 of each graph displaying no information.

The empty space was caused by a bug in the code producing the third y-axis. This is now fixed and the revised version of the manuscript will include figures without empty spaces (Fig 4-7).

**Page 14, line 19 – considering how thin the ice is here, creep closure is probably negligible in controlling the isolation of the cave atmosphere in winter. Snow plugs in the entrances are far more likely.**

We will remove the part about creep closure from the manuscript.

Page 16, Figure 9 – I don't understand the conceptual model for air flow. During summer, outside air is warmer than the cave. Air should be cooled in the upper entrance and then flow out of the lower entrance. If the cave were not plugged with snow, the opposite flow would occur in winter. I also don't understand how you have a channel in summer that lacks a stream.

This confusion is certainly caused by the round yellow circle on top of the upper entrance in Figure 9. This was indeed not meant to symbolize air flow, but heat exchange. We do agree with the way the referee describes how the air flow in the cave should work. The point we tried to make with this model, was rather to show that cave temperatures are less stable, if controlled by the air flow caused by temperature differences between surface and cave temperatures (the air flow the referee describes). Whereas they are way more stable if an additional forced convection, caused by running water, exists. The choice of figure might also be suboptimal when it comes to a cave lacking a stream during summer. The way we visualized our cave with a surface connection (upper left panel in figure 9) would almost certainly have a running stream during summer. We believe, however, that cave parts without streams do also exist during summer TCD
(please note: we are talking about cave parts, not whole caves). In case of the studied cave system at Tellbreen, this would be the case for the dead-end, where logger 'Subglacial 2' was located. At the start of the season, no connection to the surface or to other parts existed and it is therefore unlikely that water was flowing in this part of the cave, as the water entered this cave system at location D in Figure 3. However, the meltwater from the surface formed a new channel, thereby connecting the dead-end of the cave system towards the end of the melt season, thereby changing the temperature regime in this part of the cave system. We will revise Figure 9 to avoid this kind of confusion and make it more clear.

**Page 17, Figure 10 – mechanical erosion conjures images of plucking or abrasion of bedrock by suspended bed load. Perhaps fluvial erosion is a better term?**

We will change the term "mechanical erosion" to "fluvial erosion".

**Page 18, Lines 32-35 – maybe I'm missing something, but I cannot figure out what data the authors used to infer the rapid incision described here. Please clearly link this interpretation to the data.**

We derived the rapid incision from the fact, that the sediment sensors were eroded out at the end of the melt season. In the example of the Larsbreen sediment sensor, a rapid temperature increase of about 0.8°C occurred within a few days (18.06-22.06.2016), whereas no large and rapid change occurred before (see Figure S3 in the supplementary material). After this rapid temperature increase, the sediment temperature starts fluctuating similarly to the channel air temperature. We therefore conclude, that the sediment sensor was eroded out following the 22nd of June 2016. Based on the only slow temperature increase before the 18th of June 2016, which we attribute to the general warming of the channel and heat-transfer into the ground, we conclude that not much erosion occurred before the 18th. Thus, leaving the four days between the 18th and the 22nd as time frame for the main erosion. See also the response to the short comment SC1. We will detail this more in the discussion (P18, L33 onwards) and TCD
provide to additional plots in the supplementary material of the revised manuscript to make sure the reader can follow our argumentation.

**Page 19, Lines 14-34 – Much of this is extremely improbable.....**

This was also remarked by referee 1 and we will remove everything following L21 (P19. L21-34), ending the discussion at line 21 as suggested by referee 1 (see also response to RC1).
Interactive

comment

Fig. 1. Figure 4 from the manuscript with same scale for surface air and cave temperatures.

---

## Author Comment (AC3) · 30 Jul 2020

We would like to thank Yoram Terleth for reading our manuscript and providing feedback, as well as discussion input.

In the following we present our responses to the short comments and how we address these in the revision of the manuscript.

The short comments are presented in ***bold and italic***, our replies follow immediately thereafter.

**Overall comments**

[Figure]

*This very interesting study provides time series of subglacial channel temperatures and erosion under two cold based valley glaciers on Svalbard. It indicates a link between meteorological events at the surface and the yet poorly understood soft bed subglacial processes affecting glacier hydrology and potentially basal slip under cold based ice. The highlighted importance of extreme events is especially relevant in the context of understanding the effects of climatic change on Svalbard.*

We thank for this positive feedback of our study.

**Detailed comments**

*1.The study uses sudden changes in sediment temperature to identify certain erosion events, such as the August 30thevent for the Tellbreen "subglacial 1" sensor (p.18l.35) and the late July unearthing of the 1.05 m sensor under Larsbreen (p.18 l.33). However, other variations in measured sediment temperature are not addressed much in the paper, while it seems they could contribute to the compelling case for the occurrence of episodes of strong erosion linked to surface events. For example, on Figure 6 the 0.45 sediment temperature follows the channel temperature very closely from the the late June / early July peak rainfall event onward. Could it be possible that this event eroded the channel bed down quite close to the buried sensor, in addition to coupling the subglacial conduits to the atmosphere? Similarly, both 'subglacial 2' sensors in Figure 7 register a step-wise temperature increase when surface melt starts to occur around June 24th and the 0.45 sensor shows more variation after the late June rainfall. Both the 0.45 and 0.9 sensors vary with channel air temperature after the second major rainfall event of August 30th, and are exposed upon recovery. Maybe it can be argued that these observations point towards distinct episodes of stream erosion occurring over the summer season?*

Our main argument for the timing of the erosion events are not the absolute tem-

peratures, but rather the change of temperatures. In the supplementary figure S3 of the submitted manuscript, we have prepared a figure showing the change of daily subglacial temperatures compared to the previous day. Looking at the plot for the 1.05 m sensor (Fig S3 b) a small temperature increase can be seen at the start of the melt season, followed by a long stagnant period with very low temperature change compared to the previous day. We argue that this first sudden temperature increase is caused by the first water flowing into the cave system, leading to a sudden and increased heat flow into the ground. Temperatures then slowly increase until end of June (almost no daily change), speaking for slow warming through heat-exchange. Between 18th and 22nd of June we can see a rapid temperature change, followed by more fluctuating temperatures afterwards. We argue that this rapid temperature change must have been caused by erosion of the sediment and that the sensor was exposed thereafter. Due to this rapid increase within short time and almost steady temperature changes preceding to this event, we argue that most of the erosion happened within the period 18th to 22nd of June.

We have prepared similar change figures for the two Tellbreen sites 'Subglacial 1' and 'Subglacial 2'. They can be found at the end of this report. In case of the 0.45 m sediment sensor at 'Subglacial 1', almost no daily change can be observed in the period preceding the 30th of August (Fig. 1 below). The increase of the absolute temperatures that can be observed in Figure 6 is therefore rather caused by slow heat-exchange between sediment and cave air/ water. We would argue for the erosion/ re-placement at 'Subglacial 1' to happen on the 30th of August.

We argue similarly for the case of the sediment sensors at 'Subglacial 2' (see Fig. 2 below). A small temperature change of the two sediment temperatures can be seen towards the end of June and attributed to water flowing into the cave system, making heat available for heat exchange. Looking at the 0.9 m sediment sensor no further
temperature change can be seen before the 30th of August, indicating slow and steady heat-exchange, followed by fast erosion on the 30th of August. In case of the 0.45 m sensor a few episodes of temperature change can be seen in July. This could indeed be argued as a partial erosion of the sediment. It might as well also be linked to the increased heat input into the cave system as a result of the roof collapse in the lake area. As the 0.45 m sediment temperature is, however, not correlated to the surface air temperature in July (correlation coefficient -0.004, see Table 1), we argue, that the sensor itself was not exposed in July. An erosion event >0.45 m therefore occurred on the 30th of August, as both sediment sensors were exposed following this event.

We will add more explanation regarding these interpretations, and why we think a mechanism, as suggested in this comment, less likely, to the discussion part of our revised manuscript (see also response to referee 2) to make our interpretation more clear. We will further on add the two figures, provided in this response, to the supplementary material of the revised manuscript to allow the reader to better follow our interpretations.

*2.   page 19, last paragraph In the way I understand the proposed thermo-mechanical erosion mechanism (Figure10), it relies on high stream power to produce high rates of permafrost melting and erosion. The mechanism is especially effective after extreme rainfall and melt events, and applies to the channels of an efficient drainage system, which is where the measurements occurred. On lines 23 to 25, the paper mentions that a more inefficient drainage system would allow more widespread influence of extreme events on basal slip. It would be nice to clarify what is meant exactly, as it seems that in a fully inefficient distributed drainage system, water flow velocities would be too low to allow for turbulent heating and the thermo-mechanical erosion mechanism to occur. It could be interesting to consider Rippin et al. (2005), as they suggest that after mass build-*

*up, pressurized and fast water flow through the cold based margin sediments could increase local ice velocities. This seems like it could be a situation where the mechanism presented in this study would be quite relevant.*

While it is certainly correct, that in most cases an inefficient drainage system means low flow velocities and thus low erosional stream power, it does not exclude the possibility for high water flow velocities (as Yoram Terleth also mentions in regards to Rippin et al. (2005)) and turbulent flow. For more information see Flowers (2005) and the reference therein to Alley (1996), where turbulent flow is addressed for models of inefficient drainage. We will, however, remove this paragraph from the revised manuscript (P19, L21-34) as suggested by referee 1 and referee 2 (see RC1 and RC2, as well as author responses to both reviews).

References:
Alley RB. (1996) Toward a hydrologic model for computerized ice-sheet simulations. Hydrol. Proc. 10, 649–660.

Flowers Gwenn E. (2015). Modelling water flow under glaciers and ice sheets. Proceedings of the Royal Society A: Mathematical, Physical and Engineering Sciences 471, 20140907.

*3.A final short remark is that in a recent paper, Haga et al. (2020) mention the potential importance of an efficient drainage system in the partial freezing of the Negribreen glacier terminus to its bed surface. The rapid erosion in response to surface events in this study could indicate the capacity of a drainage system to adjust rapidly to changes, even in permafrost. Maybe such an adjustable system is necessary for the cold based conditions of many Svalbard glaciers termini to form, or at least facilitates formation?*

The data presented in our study does indeed indicate a fast adjustment possibility for efficient drainage channels in permafrost, given enough available subglacial sediment.

A surge-type glacier such as Negribreen would certainly provide enough sediment for rapid channel adjustment, with the limitations outlined by referee 2 in his first detailed comment (see RC2). The cold based conditions of many Svalbard glacier termini are, however, more likely caused by a combination of thin ice and cold winter temperatures, leading to a cooling of ice and underlying sediment (if the ice is thin enough). This would especially be the case for Negribreen, which is located at the East coast of Svalbard, which is considerable colder than the west coast, due to ocean currents.
* * *
[Figure]

**Fig. 1.** Change of daily temperatures at logger 'Subglacial 1'.

**(a) Channel air temperature change (°C)**

Rainfall (mm)
Surface air temperature (°C)
Channel air temperature change (°C)

**(b) 0.45 m sediment temperature change (°C)**

Rainfall (mm)
Surface air temperature (°C)
0.45 m sediment temperature change (°C)

**(c) 0.9 m sediment temperature change (°C)**

Rainfall (mm)
Surface air temperature (°C)
0.9 m sediment temperature change (°C)

**(d) 0.9 m ice temperature change (°C)**

Rainfall (mm)
Surface air temperature (°C)
0.9 m ice temperature change (°C)

**Fig. 2.** Change of daily temperatures at logger 'Subglacial 2'.

---

## Author Response (AR1)

**Author response after major revision: Surface event driven subglacial permafrost dynamics and erosion inside subglacial channels**

Andreas Alexander[1,2], Jaroslav Obu[1], Thomas V. Schuler[1], Andreas Kääb[1], Hanne H. Christiansen[2]

[1]Department of Geosciences, University of Oslo, 0316 Oslo, Norway
[2]Department of Arctic Geology, The University Centre in Svalbard, 9171 Longyearbyen, Norway

*Correspondence to*: Andreas Alexander (andreas.alexander@geo.uio.no)

We would like to thank Andreas Vieli, Doug Benn and one anonymous referee for going through our manuscript and providing very helpful feedback, that clearly improved the quality of our manuscript. We have addressed all referee comments (see detailed answers below), updated the figures and an additional figure to explain the logger setup (figure 5). We have further added more modern literature to the introduction and method section of the manuscript and improved the discussion. A mark-up version of the manuscript, showing the changes made in response to the referee's comments, can be found at the end of this document.

**Response to the review by Doug Benn**

First of all we would like to thank Doug Benn for reviewing our manuscript and provide helpful and constructive feedback, which certainly helped to improve the paper.
In the following we present our responses to the referee comments and how we addressed these in the revision of the manuscript.

The referee comments are presented in **bold and italic**, our replies follow immediately thereafter.

**Overall comments**

***This is a worthwhile paper, that presents novel and useful data on air and ground temperatures within subglacial conduits under a cold-based glacier tongue in Svalbard. The data include good cave maps and very useful temperature series from a number of sites, spanning both warm and cold parts of the year. The paper is very clearly written and structured, and most is ready for publication without revision.***

We thank the reviewer for this overall positive judgment of our manuscript.

*The only shortcomings with the paper concern how it is placed in the context of previous work, and the significance of the some of the conclusions, which is rather over-stated in the closing paragraph of the Discussion.*

We thank the reviewer for pointing out the shortcomings of our initial manuscript in regards to the literature. We have improved the introduction of the manuscript with the suggestions from Doug Benn, as well as the second anonymous reviewer (P2, L24-34). We have further on removed the scrutinized closing paragraph of the discussion part from the manuscript (P19 lines 21-34 of the initial manuscript).

**Detailed comments**

*Page 2, Line 26: high sedimentation rates in polythermal and cold-based glacier catchments. The authors have missed the most important factor concerning the sediment budget of these glaciers: most of the examples cited are either surge-type or were more dynamically active during the Little Ice Age.*

We thank the reviewer for pointing out this factor. We have indeed missed it.

*The Hodgkins study focused on Finsterwalder-breen (surge-type); in Hallet's global compilation the Svalbard examples are surge-type; Etzelmüller looked at Larsbreen and Longyearbreen, both of which were more dynamically active in the past (see Sevestre et al., 2015 regarding former dynamics of Longyearbreen).*

While we see the reviewers point regarding the Hodgkins and Hallet studies, we still consider Etzelmüllers' study of Larsbreen and Longyearbreen as relevant literature, as Larsbreen is one of the two study sites of this manuscript.

*Sollid and Sørbel conducted a palaeo-study and inferred the glacier thermal regime, so this does not provide independent evidence of the link between thermal regime and sediment dynamics. My point is that the high sediment load on Svalbard glaciers mostly relates to past surges, in which sediment can be elevated to high levels by thrusting and other processes. This sediment is then released and reworked by fluvial and gravitational processes during quiescence. Papers by Lovell should be cited in this respect. (e.g. 1: Lovell, H., Fleming, E.J., Benn, D.I., Hubbard,B., Lukas, S. and Naegeli, K., 2015. Former dynamic behaviour of a cold-based valley glacier on Svalbard revealed by basal ice and structural glaciology investigations. Journal of Glaciology, 61(226), pp.309-328. 2: Lovell, H., Benn, D.I., Lukas, S., Otte-sen, D., Luckman, A., Hardiman, M., Barr, I.D., Boston, C.M. and Sevestre, H., 2018.Multiple Late Holocene surges of a High-Arctic tidewater glacier system in Svalbard. Quaternary Science Reviews, 201, pp.162-185. 3: Lovell, H., Fleming, E.J., Benn, D.I.,Hubbard, B., Lukas, S., Rea, B.R., Noormets, R. and Flink, A.E., 2015. Debris entrainment and landform genesis during tidewater glacier surges. Journal of Geophysical Research: Earth Surface, 120(8), pp.1574-1595.).*

*Lines 28-9: This statement implies that there is a 'missing' process of sediment erosion. This is not the case. Boulton (1972) is a very old source with regard to*

*erosion mechanisms; much more recent and comprehensive sources can be cited, which give more attention to fluvial processes. Additionally, the erosional capability of sub-glacial channels under cold glaciers (and Tellbreen in particular) was flagged up by Naegeli et al. 2014. Dendritic subglacial drainage systems in cold glaciers formed bycută˘AˇRandă˘AˇRclosure processes. Geografiska Annaler: Series A, Physical Geogra-phy, 96(4), pp.591-608.*

We have removed the scrutinized paragraph from the introduction and have accounted for more recent literature (P2, L24-34).

*Page 4, line 9: Recent work on Tellbreen should be cited here, to provide proper context for the study. Key facts from the following papers should be summarised in a sentence or two at this point in the paper: Origin of the subglacial channels in Tellbreen: Naegeli et al. 2014 (cited above), and in other Svalbard glaciers: Gulley, J.D., Benn, D.I., Müller, D. and Luckman, A., 2009.A cut-and-closure origin for englacial conduits in uncrevassed regions of polythermalglaciers. Journal of Glaciology, 55(189), pp.66-80.Dynamical history of Tellbreen: Lovell, H., Fleming, E.J., Benn, D.I., Hubbard, B.,Lukas, S. and Naegeli, K., 2015. Former dynamic behaviour of a cold-based val-ley glacier on Svalbard revealed by basal ice and structural glaciology investigations. Journal of Glaciology, 61(226), pp.309-328.*

We have added the reference to Lovell et al. 2015 (P6, L1), as well as the reference to Naegeli et al. 2014 and Gulley et al. 2009 (P6, L5-6).

*Most of the Discussion is well written, building a set of sound conclusions and inferences from the data. The sections on subglacial channel erosion are especially welcome. This process has been previously inferred from the existence of subglacial channels at Tellbreen (Naegeli et al.), but the present paper adds valuable insights about processes and rates.*

We thank the reviewer for this positive feedback.

*However, two points in the Discussion need attention:*
*Page 19, line 3-6. It is difficult to see how refreezing of the sediment should cause such a catastrophic drop in sensor temperature. A phase change from liquid to solid in surrounding saturated sediment should result in a temperature increase, not a drop, because freezing gives up latent heat. This feature of the record almost certainly reflects sensor malfunction.*

We have double-checked in the non-aggregated data (one minute resolution versus daily) and there is a slow temperature increase visible, shortly before the catastrophic temperature drop. We therefore attribute this catastrophic temperature drop to a combination of re-positioning of the sensor in the sediment, followed by refreezing of the sediment, leading to release of latent heat (increase of sensor temperature), followed by further freezing and strain-induced sensor malfunction caused by the refreezing. This has also likely lead to the complete failure of the second sensor and the

sensor cables were clearly under tension, during data recovery in autumn 2019. We have added according information to the discussion text (P21, L1-6) of the manuscript and added an additional figure to the supplementary material of the manuscript (figure S6).

*Page 19, lines 21-34. From this point on, the Discussion loses its grip on reality and wanders off into wild speculation. Beneath thinning Svalbard glaciers, the thermal trend is from warm to cold-based conditions, as diminishing ice thickness allows conductive losses to the surface to increase during winter. The authors have convincingly shown that this trend can be reversed locally by the presence of channels which advect additional heat to the bed from the surface during the summer months. There is nothing in the data that indicate that these highly localised and seasonal changes could impact the broader hydrological system or dynamics. Indeed, Tellbreen, like the majority of small glaciers in Svalbard, has strongly negative surface mass balance and is in terminal decline. The trends of thinning ice and permafrost aggradation will continue regardless of local seasonal heating around surface-fed conduits. The paper does not need vague speculation about wider 'impact' in order to be relevant - indeed, the paper is weakened by it. Just end the Discussion at line 21.*

We have removed everything, following line 21 of the initial manuscript.

**Response to RC2**

We would like to thank the anonymous referee for taking the time to review our manuscript and for providing thorough and helpful feedback, which helped to improve the quality of the manuscript.

In the following we present our responses to the referee comments and how we addressed them in the revised version of the manuscript.

The referee comments are presented in **bold and italic**, our replies follow immediately thereafter.

**Overall comments**

***This manuscript provides new information concerning how temperature variability in subglacial channels can impact fluvial erosion beneath cold-based glaciers in Svalbard. I'm not aware of a similar dataset and the results should be of interest to a broad community of glacier and permafrost researchers.***

We thank the referee for this overall positive judgment of our dataset.

***While I think the paper should be published eventually, I'd like to see the authors more closely situate their manuscript within the modern published literature on the sedimentology and hydrology of cold ice glaciers in Svalbard, provide a clearer description of the sensor installations in the methods section, and more clearly link the discussion to their results. More detailed comments are included with page and line numbers below.***

We thank the referee for his feedback. We have reworked the introduction of our manuscript to incorporate more modern literature, as requested both by referee 1 and 2 following their suggestions (see also response to RC1). We have further on clarified the sensor installations in the methods section and improved the discussion part following the suggestions from both referees.

**Detailed comments**

***Page 1, Lines 15-20 – Fluvial incision of subglacial tills can erode sediment, but vertical incision of subglacial channels can become limited by boulder armoring. Fine grained materials are preferentially winnowed from till channel by flow and larger boulders and rocks accumulate on the floor (See Gulley et al., 2014). Because flow cannot mobilize these sediments, vertical incision largely ceases but the channel can still migrate and incise laterally. In the case of the till beneath cold-***

*based glaciers in Svalbard, much of the sediment being eroded by streams was not produced beneath cold based glaciers, as seems to be implied by the authors, but instead is derived from past temperate basal regime or perhaps surges.*

We have amended the abstract to point out, that only available sediment can be eroded (P1, L19). We have further on included the part of winnowing of fine-grained material and accumulation of larger blocks at the channel floor, together with the reference to Gulley et al., 2014, in the introduction of the manuscript (P2, 33-34).

*Page 2, Lines 25 – Boulton 1972 was written when it was widely believed that cold-based glaciers lacked active subglacial hydrological systems. Hodgkins (1997) highlighted water flow and erosion beneath cold ice in Svalbard. More recent work has shown that much of the water flowing out from under cold-based glaciers is likely de-rived from subglacial channels which began life as supraglacial streams and later incised through cold ice to reach glacier beds (see Gulley et al., 2009).*

We have removed the scrutinized paragraph from the introduction and have accounted for more recent literature (P2, L24-34).

*Page 4, Figure 1b and c – drawing an outline of the subglacial channel beneath Tellbreen in one panel and only showing the survey line of the channel beneath Larsbreen in the other panel is confusing. At first glance, the cave beneath Tellbreen looks like it is a giant loop! Also, "profile" is typically used to describe what the authors are calling "vertical cross-section (extended profile)" and cross-section is typically used to describe the cross-section of a passage segment (drawn perpendicular to the passage direction).*

We have updated the map in figure 1, as well as the caption of figure 1 to implement the reviewers' comments. We have further on also updated the terminology in figure 2 and 3.

*Page 4, Line 7 – remove "is reported"*

The removal of this part would render the sentence incomplete (missing verb), we have therefore left it in.

*Pages 5,6 - Figures 2 and 3 – there is too much information jammed into these figures. I recommend separating the maps from the photo panels and making the photo panels larger. The pictures are too small to see the information being described in them. For example, there is a logger setup mentioned in Fig 3C – but I can't see it.*

We have separated the pictures from the maps (figure 2-4) to increase the readability and have formulated the figure captions more clearly, as the logger setup was in figure

3D not 3C of the initial manuscript. There is, however, a limitation to which the initial size of the pictures can be enlarged without blowing up the manuscript.

***Methods – HOBO pendant loggers have an accuracy of 0.53 deg C when used forabove-freezing temperatures. Accuracy decreases to 0.75 deg C between 0 and -20Chttps://www.onsetcomp.com/products/data-loggers/ua-002-64/***

We did not use the model indicated by the referee, but the Pendant MX version: [https://www.onsetcomp.com/products/data-loggers/mx2202/](https://www.onsetcomp.com/products/data-loggers/mx2202/)

We have added this information to the manuscript and corrected the given resolution, as there was one digit too much (P9, L2).

***Page 7, line 20 and 21 – I don't understand this installation. Was the pipe filled with silicone? If so, please make it clear why. Why were the temperature sensors not buried directly in sediment? For the logger installed into the ice, was it also installed in pvc pipe? What diameter holes were drilled in the ice and sediment and what was the diameter of the pipe? A clear line drawing showing the various types of instrumentation as installed in the caves would be very helpful in visualizing the experiment and this information is critical for interpreting the temperature signals.***

We have added an additional figure to the manuscript to explain the logger setup at the two glaciers (figure 5). Further on, we have added information to the text describing the logger setup to clarify the questions raised by the reviewer (P8, L6-9 and P9, L8-14).

***Page 7, Line 30 – please describe the calibration procedure used for the DistoX.***

We added according information to the manuscript (P9, L23-24).

***Page 9, Fig 4 – It would really help visualize relationships between outside air temperature and the cave/sediment temperatures if they were plotted at the same scale.***

We produced such a figure, see the below figure. As the cave temperatures appear at the scale of the outside temperatures almost as a straight line without much information given, we decided to leave the temperatures in Fig 6 at different scales to allow for easier interpretation of the results.

[Figure]

*Page 14, line 5 – the authors keep referring to "winter" but the dataset only runs from March until October. It would be nice if the time series graphs could be truncated at the end of the data collection period instead of having 1/5 of each graph displaying no information.*

We have updated Fig 6-9 accordingly.

*Page 14, line 19 – considering how thin the ice is here, creep closure is probably negligible in controlling the isolation of the cave atmosphere in winter. Snow plugs in the entrances are far more likely.*

We have removed the part about creep closure from the manuscript

*Page 16, Figure 9 – I don't understand the conceptual model for air flow. During summer, outside air is warmer than the cave. Air should be cooled in the upper entrance and then flow out of the lower entrance. If the cave were not plugged with snow, the opposite flow would occur in winter. I also don't understand how you have a channel in summer that lacks a stream.*

We have updated figure 11 to make it clearer.

*Page 17, Figure 10 – mechanical erosion conjures images of plucking or abrasion of bedrock by suspended bed load. Perhaps fluvial erosion is a better term?*

We have updated Fig. 12 accordingly.

*Page 18, Lines 32-35 – maybe I'm missing something, but I cannot figure out what data the authors used to infer the rapid incision described here. Please clearly link this interpretation to the data.*

We have added additional information to the discussion text of the manuscript to clarify how we inferred the rapid incision (P20, L25-29) and added additional figures in the supplement that support our interpretation (S4-5).

*Page 19, Lines 14-34 – Much of this is extremely improbable . . ..*

We have removed everything following Line 21 of the initial manuscript.

[revised manuscript text omitted]
 | 0.35 | 0.01 | −0.10 | 0.36 | −0.12 | −0.35 | 0.40 | −0.49 | 0.23 | 0.16 | 0.09 | |

---

## Author Response (AR2)

**Author response after minor revision: Surface event driven subglacial permafrost dynamics and erosion inside subglacial channels**

Andreas Alexander[1,2], Jaroslav Obu[1], Thomas V. Schuler[1], Andreas Kääb[1], Hanne H. Christiansen[2]

[1]Department of Geosciences, University of Oslo, 0316 Oslo, Norway
[2]Department of Arctic Geology, The University Centre in Svalbard, 9171 Longyearbyen, Norway

*Correspondence to*: Andreas Alexander (andreas.alexander@geo.uio.no)

We would again like to express our thanks to the editor and the two referees who helped to improve the quality of the manuscript and helped to get it ready for publication. We have followed the final suggestions/ requests from the editor and provide our answer to his points below. We have further uploaded the raw data from our study to the NIRD research data archive and will provide the doi for the dataset as soon as we have obtained it.
A mark-up version of the manuscript showing the changes is following at the end of the document.

**Response to minor corrections/ editing issues from the editor**

The editor comments are presented in ***bold and italic***, our replies follow immediately thereafter.

***Fig. 5: '...the sediment temperature SENSOR was buried...' (the 'sensor' was missing).***

Thanks for catching this error. We fixed it.

***Fig. 6-9: I would find it useful to add a zero degree horizontal orange line, for the right orange scale. I accept that the authors do not want to use just one temp. scales, but a zero line would help as a reference.***

That's a very good suggestion and we have added the zero lines to fig 6-9.

***Fig. 11: I struggle a bit with this conceptual figure: i) why is there no mouling chabnnel in. the top left summer figure? Is this to illustrate the case of no subglacial waterflow? But why should this be, I assume it needs water to have a channel there ii) related, is a subglacial channel without water flowing realistic in summer (see comment referee 2), iii) it would be quite useful to see what temperature regime there is (where is it cold/warm...) and potentially in which direction the air is expected to flow (see also comments by referee 2).***

***Anyway, this figure should be clarified a bit, perhaps just describe the 4 situation (and differences) clearly in the caption***.

This figure does indeed leave room for interpretation and misunderstanding.

i) We have added the moulin again to this part of the figure. We have also added water flow following in the channel after the moulin.

ii) Our observations show, however, that under thin ice subglacial channel sections without water flow exist. These sections form by the water cutting new pathways, thereby leaving dead-ends (similar dead-end formation can be observed in supraglacial channels). As the ice overburden pressure is not high enough, these channel sections remain open for several years.
We have added additional information in the figure and the figure caption to explain this better.

iii) The figure is not intended to show the temperatures itself (this would anyways be quite hard), but rather the controlling factors on the temperature. We therefore, choose to not show color-coded temperatures as this would introduce another level of potential misinterpretation. We also choose, for a similar reason, to not represent the expected air flow from natural convection, but rather only show the arrows for forced convection.

iv) We have re-written the figure caption to better address the points raised by the editor and hopefully contribute to an easier understanding of the figure itself.

[revised manuscript text omitted]